# Thresholds for adding degraded tropical forest to the conservation estate

Logged and disturbed forests are often viewed as degraded and depauperate environments compared with primary forest. However, they are dynamic ecosystems[1] that provide refugia for large amounts of biodiversity[2,3], so we cannot afford to underestimate their conservation value[4]. Here we present empirically defined thresholds for categorizing the conservation value of logged forests, using one of the most comprehensive assessments of taxon responses to habitat degradation in any tropical forest environment. We analysed the impact of logging intensity on the individual occurrence patterns of 1,681 taxa belonging to 86 taxonomic orders and 126 functional groups in Sabah, Malaysia. Our results demonstrate the existence of two conservation-relevant thresholds. First, lightly logged forests (<29% biomass removal) retain high conservation value and a largely intact functional composition, and are therefore likely to recover their pre-logging values if allowed to undergo natural regeneration. Second, the most extreme impacts occur in heavily degraded forests with more than two-thirds (>68%) of their biomass removed, and these are likely to require more expensive measures to recover their biodiversity value. Overall, our data confirm that primary forests are irreplaceable[5], but they also reinforce the message that logged forests retain considerable conservation value that should not be overlooked.

Habitat degradation has seemingly contradictory impacts on the biodiversity of tropical forests. Human disturbance of tropical forests has resulted in the same amount of biodiversity loss as outright deforestation[6], leading to a widespread view that logged, degraded and regenerating tropical rainforests are depauperate environments relative to primary forest[5]. However, logged forests are also more dynamic environments than primary forest[1], can have elevated habitat heterogeneity[7], support enhanced populations of many taxa[8] and provide refugia for a remarkable diversity of species[2,3]. Given this apparent paradox, it is not immediately clear whether degraded forests should be considered as conservation assets or not. As logged forests increasingly dominate tropical landscapes[9,10], questions around their conservation protection should be a priority. The intensity of logging varies greatly within and among tropical regions[11,12], which further complicates the debate around the conservation of logged and degraded forests. Precedents exist of even heavily logged forest being afforded the strictest levels of conservation protection[13], but we lack clear evidence about whether this approach should be expanded.

Conservation actions can be largely categorized as being either proactive or reactive[14]. Proactive conservation targets areas of low vulnerability, where approaches such as protecting the habitat are expected to deliver positive outcomes for biodiversity. By contrast, reactive conservation targets areas of high threat, where immediate action is required to stave off biodiversity loss. Lightly logged forest might retain sufficient biodiversity and ecological value to justify formal conservation protection, should that be a socially equitable approach in the region of interest[15]. This proactive approach to conservation in largely intact ecosystems seeks primarily to prevent additional habitat degradation from taking place. However, more heavily degraded forests might also require costly reactive conservation interventions—such as remediation, restoration and long-term management[14]—to accompany the protection of the habitat. In this study, we quantify how much damage a forest can sustain before proactive conservation approaches might need to be replaced with reactive approaches, identifying two ecological thresholds that can be used to guide conservation decisions of this nature.

Identifying thresholds requires the quantification of biodiversity responses to disturbances, such as logging in tropical forests[16], which seems deceptively simple. Hundreds, if not thousands, of individual empirical studies have tackled this question, but each is commonly limited to one or a small number of taxonomic groups such as plants[17], mammals[18], birds[19] or ants[20], which creates two challenges. First, responses to forest degradation are often taxon-specific[21,22], although there are some landscape-level thresholds in community responses that exhibit remarkable congruence[23]. Second, taxon-specific studies can easily exaggerate perceived impacts on ecological functions, because they are unable to capture compensation by functionally similar taxa in unrelated taxonomic groups[24,25]. Consequently, answers obtained from taxonomically limited studies can reflect the researchers' choice of study taxa more than the community-wide effects of degradation on biodiversity and ecosystem functioning. This confusion of taxon-specific responses and cross-taxon ecological redundancy means that we have little synthetic understanding of where to target different forms of conservation action along gradients of habitat degradation.

Here, we surmount these challenges by summarizing responses collated across 127 biodiversity surveys (Supplementary Table 1). Each survey took place in a single year, and all were conducted during an 11-year period at the Stability of Altered Forest Ecosystems Project in Sabah, Malaysia[26,27]. This experimental landscape encompasses a continuous gradient in logging intensity that ranges from unlogged

primary forest, through salvage-logged forest (where no limits were placed on the number or size of trees to be removed), to riparian forest in protected riverine buffer zones and forest converted into oil palm plantations. Along this gradient, the percentage of biomass removed varied from 0 to 99%, which we use as a generalized metric of forest degradation. This metric implicitly combines the initial removal of woody biomass through one or more logging and land clearance events with the gradual recovery of biomass that may have occurred since the last disturbance event(s), meaning that our metric of forest degradation reflects the present-day balance between these two opposing forces. From previous work at this site, we have shown that forest degradation causes changes to local environmental conditions, including the microclimate[28] and the functional composition of the tree community[29].

Together, the biodiversity surveys contain information on the occurrence patterns of 4,689 terrestrial and aquatic taxa (Extended Data Fig. 1) and 126 functional and morphological groups (Methods and Supplementary Table 2). Of these, 1,681 taxa and all 126 functional groups were observed ≥5 times and were able to be modelled individually. Of the 1,681 taxa we modelled, more than half (n = 946, 56%) were detected in more than 1 survey (Extended Data Fig. 2), and more than half (54%) of individual surveys consisted of multiple site visits (repeated observations of the same sites within the survey year). The taxa were widely distributed across the tree of life (Extended Data Fig. 1) and encompassed representatives from 86 taxonomic orders and 679 genera, including 590 plants (understorey and canopy, including grasses, herbs and woody trees), 88 mammals (including bats), 161 birds, 9 reptiles, 42 amphibians, 26 fish and 635 invertebrates (including 263 beetles, 199 lepidopterans, 130 ants and 33 spiders). The taxa ranged in body size over 8 orders of magnitude from the smallest featherwing beetles in the family Ptiliidae (17 mg) to the Bornean elephant *Elephas maximus* (3.2 tonnes), encompassed 21 diet groups spread across 6 trophic levels, and represented 18 categories of movement mode, physiology, habitat use, sociality and conservation status (Methods and Supplementary Table 2). Functional groups based on trophic levels and diet were agnostic to taxonomy, recognizing for example that both spiders and birds have insect prey and can contribute to the same ecological function[25] (Methods).

We focus our analyses on two critical points in the responses of individual taxa to habitat degradation. We define a 'change point' as the first point along the degradation gradient at which a taxon exhibits a discernible change in occurrence probability. We then define a 'maximum rate point', which represents the point along the forest degradation gradient where the rate of change in occurrence probability is the most rapid. Both change and maximum rate points were calculated from derivatives of fitted occurrence models (Methods and Extended Data Fig. 3).

## Degradation has an immediate impact

No level of forest degradation was too low to have an impact (Fig. 1a): the occurrence patterns of 24% (n = 396) of taxa and 34% (n = 41) of functional groups were affected from the onset of biomass removal. Although seemingly extreme, such intense sensitivity to small amounts of forest disturbance echoes earlier, global analyses showing that tropical taxa in intact habitats are heavily affected by very small amounts of forest loss[30].

More taxa and functional groups were negatively (425 and 51, respectively) than positively (330 and 32) affected by forest degradation, so the mean occurrence level reduced slowly as forest degradation increased (Fig. 1b). Remarkably, the 811 taxa that were present in unlogged forest (≤5% biomass removal) were twice as likely to have positive (28%, n = 228 taxa) than negative (14%, n = 110) responses to forest degradation, which reinforces previous analyses showing how logged forests have higher ecosystem energy flows and higher species

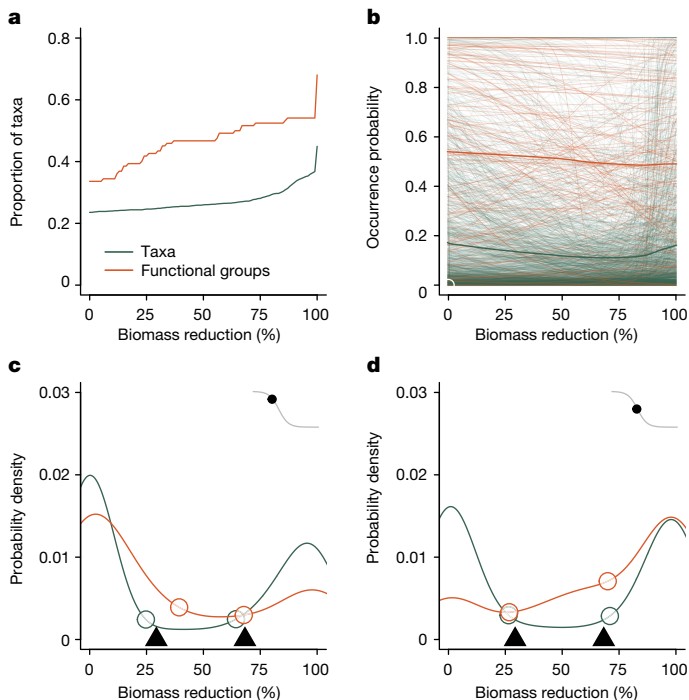

**Fig. 1 | Summarized responses of 1,681 taxa and 126 functional groups to forest degradation.** Forest degradation is represented as a percentage reduction in above-ground biomass, for which zero represents the median biomass in unlogged forest. **a**, Cumulative distribution function of the proportion of taxa or functional groups that have passed a change point along the forest degradation gradient. **b**, Mean occurrence probabilities along the forest degradation gradient. Thin lines show the fitted lines for all individual taxa and functional groups. Thick lines show the unweighted mean value of all fitted lines. **c**,**d**, Probability distribution functions showing the spread of change points (**c**) and maximum rate points (**d**) in occurrence for individual taxa and functional groups. Insets present a stylized representation of how change and maximum rate points are identified (see Extended Data Fig. 3 for a more detailed explanation). Open circles represent locations at which the rate of accumulation of taxa accelerates, and are used to estimate thresholds (filled triangles) for conservation action (Methods). Peaks in the distributions represent points along the degradation gradient where the largest number of taxa or functional groups begin to be first affected (**c**) or have their maximum rate of change in occurrence probability (**d**).

richness than primary forest[1]. However, we emphasize that those taxa and functional groups that directly benefit from logging—around one-fifth of the study taxa—do not necessarily mitigate losses in other taxa: any human-caused change in the ecosystem, whether positive or negative for an individual taxon or functional group, is noteworthy and potentially concerning. Increased occurrence can be a positive outcome for a specific taxon, yet represent a negative outcome for the ecosystem if, for example, they are invasive species. Forest degradation at our study site has promoted the invasion of non-native rodents[31] and plants[32], which is a globally common pattern[33]. However, there are many native and endemic taxa that do benefit from forest degradation, including invertebrate, bird and mammal species[1,2,8,24] that can exploit the higher bottom-up provision of food resources such as fruits[24] and more palatable foliage[1] in degraded forests. Our study site also has low hunting pressure when compared to other logged forests in the wider region[34,35], so may represent a more positive outcome than expected in comparably degraded forests with more hunting. Nonetheless, it is clear that if hunting is restricted, logged and degraded forests can support high biodiversity and ecological value[35].

Many taxa and functional groups had change points (Fig. 1c) at low levels of biomass removal, and a maximum rate of change (Fig. 1d) in

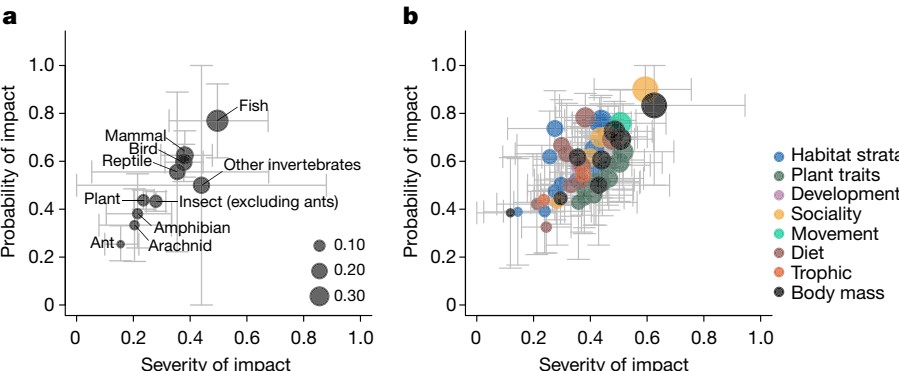

**Fig. 2 | Vulnerability of 10 taxonomic groups and 47 functional traits to habitat degradation. a**, Taxonomic vulnerability. **b**, Functional vulnerability. The magnitude of vulnerability is indicated by the size of the plotted points, and is the product of metrics representing the probability and severity of impact that habitat degradation has on taxa within the groups (Methods). Probability of impact is represented as the proportion of individual taxa within the group that had statistically significant changes in occurrence along the forest degradation gradient. Severity of impact is calculated as one minus the mean proportion of biomass reduction at which individual taxa within the group have change points. Points are plotted at the mean values of probability and severity of impact per group, and whiskers represent the bootstrapped 95% confidence interval. We assigned 1,681 taxa to 1 of 10 taxonomic groups (**a**), and to all of the 126 functional traits for which those taxa exhibited matching characteristics (**b**; Methods and Supplementary Table 2). Only functional traits containing ≥5 taxa are shown (*n* = 47). Functional groups are coloured according to broadly defined functional categories.

only lightly degraded forest. Together, these two patterns reinforce the unique and irreplaceable value of unlogged forest habitat[5]. Low-intensity logging of forests continued to affect additional taxa and functional groups until around 30% of biomass had been removed (Fig. 1c), after which more severe logging exerted little additional influence on the occurrence patterns of taxa until approximately 80% of biomass had been removed. Past this latter point, the act of removing the last remaining trees began to rapidly affect a new suite of taxa and functional groups (Fig. 1a).

## Ecological thresholds for conservation

Our results indicate that forest that has lost less than 29% of biomass (95% bootstrapped confidence interval = 25–35%; Methods) is likely to retain relatively high biodiversity and ecological value, and should be considered a viable addition to the proactively managed conservation estate (Fig. 1c,d). This value is similar to the more arbitrary definition of a high-density forest in the widely used High Carbon Stock approach[36], which sets an arbitrary threshold at 150 t ha$^{-1}$ of carbon, regardless of pre-logging biomass (equivalent to 25% biomass reduction at our study site). Our threshold value represents the point at which changes in the occurrence patterns of many taxa have taken place (Fig. 1c), and where the number of functional groups experiencing maximum rates of change in occurrence begins to accelerate (Fig. 1d). However, most functional groups have had only relatively small changes in occurrence patterns at the 29% threshold, implying that the forest retains strong potential to recover through natural secondary successional processes if left alone, and means that its conservation value can confidently be expected to increase through time without requiring direct, and often costly, management interventions.

Reactive conservation action may be best targeted in extremely degraded forests with around two-thirds of biomass loss (Fig. 1c,d; 68%, 95% bootstrapped confidence interval 60–83%). Change points represent early signals of impending ecological changes, but those impending changes, and by association the largest ecological impacts, will begin to fully manifest only as taxa and functional groups reach their maximum rates of change in occurrence. The number of taxa and functional groups reaching maximum rates accelerated rapidly after 70% biomass reduction (Fig. 1d). Even small improvements to the condition of the forest in this portion of the degradation gradient may be expected to have large impacts on the occurrence patterns of both individual taxa and functional groups, suggesting that remedial action such as underplanting or liana cutting will probably be most effective if targeted here. We note, however, that our analysis examines the directed transition from unlogged to logged forest, and that our threshold is unlikely to mark the point at which taxa and functional groups recover to the same level following restoration of logged forest: a higher level of biomass restoration is probably required[37].

## Vulnerability to forest degradation

Forest degradation affected taxa across the tree of life, but unevenly (Fig. 2a), emphasizing how answers to critical conservation questions can be dependent on choices of study taxon. Of the 86 taxonomic orders in our analysis, 72 (81%) included taxa whose occurrence patterns were significantly altered by habitat degradation, as were those of 83 (68%) of the functional groups we analysed. We calculated the vulnerability of taxonomic and functional groups to habitat degradation as the product of probability of impact (the proportion of taxa within that group that were significantly affected), and severity of impact (mean location of change points along the forest degradation gradient; Methods). Both taxonomic and functional groups containing taxa that have a high probability of being affected also tended to have a high severity of impact (Fig. 2, Pearson correlation, taxonomic groups: $r = 0.92$, d.f. = 8, $P < 0.001$; functional groups: $r = 0.69$, d.f. = 45, $P < 0.001$).

Across the major taxonomic groups, vertebrates were more vulnerable than invertebrates. Fish were the group with the highest proportion of taxa that were significantly affected (77%), and one of the most severely affected groups, with many taxa heavily affected by the early onset of logging[31]. Consequently, fish were also the most vulnerable taxonomic group to forest degradation, whereas ants and arachnids were the least vulnerable (Fig. 2a). Mammals also had high vulnerability to logging, which corroborates a previous pantropical analysis[11].

We found no significant differences in vulnerability among the different functional trait categories in the analysis (Fig. 2b, $\beta$-regression; $\chi_7^2 = 7.77$, $P = 0.35$). Rather, most categories of traits exhibited a range of vulnerability, reflecting the tremendous amount of variation of specific traits nested within those categories (Fig. 3a). For example, understorey birds had high vulnerability whereas arboreal mammals had low vulnerability, yet both functional groups represent traits related to the habitat strata they occupy and were therefore grouped together for this analysis.

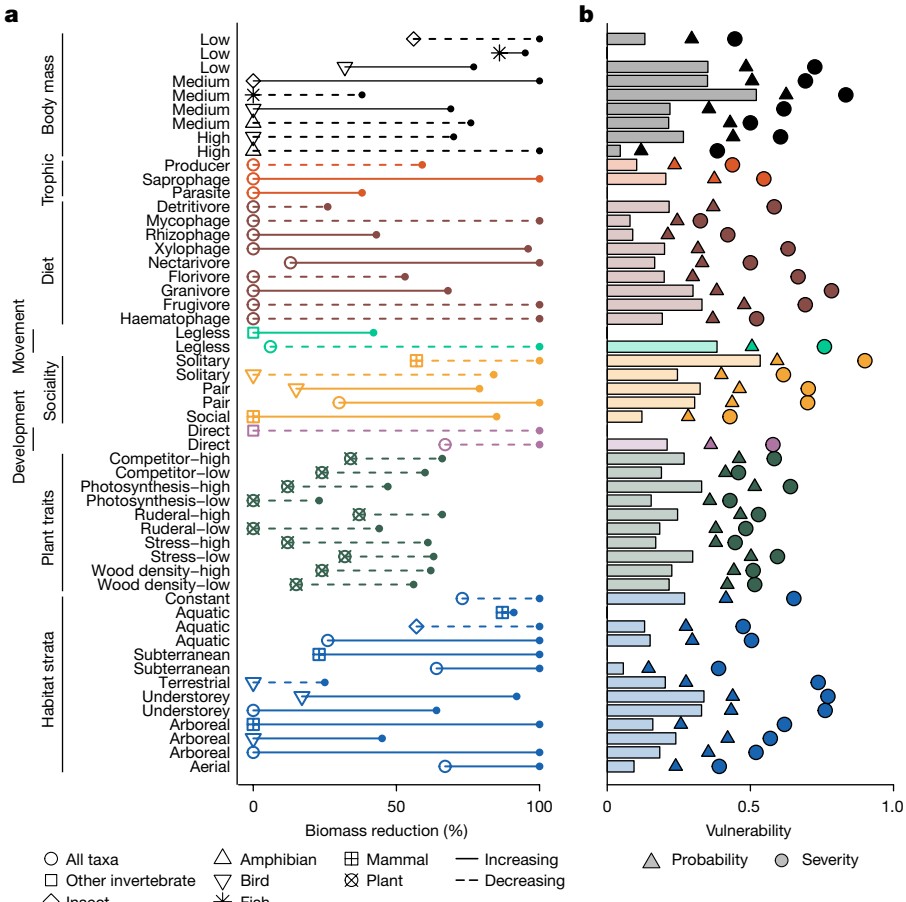

**Fig. 3 | Functional group responses to a forest degradation gradient.**
**a**,**b**, Data show the impact of biomass reduction on critical thresholds and turnover (**a**), and vulnerability, probability of impact and severity of impact (**b**) of functional groups. Analyses were conducted on the 126 functional groups described in Supplementary Table 2, but here we present only functional groups that had statistically significant responses to forest degradation. All other groups not shown had non-significant responses. In **a**, lines represent a single functional group and connect the change point (symbol) to the maximum rate point (dot) for that group. Line type indicates whether the occurrence probability of that functional group is increasing (solid) or decreasing (dashed) along the forest degradation gradient, and symbols represent different taxa. The 'Other invertebrate' grouping contains non-insect invertebrates. In **b**, vulnerability is shown in bars, with symbols representing the probability and severity of the impact that habitat degradation has on taxa within the groups. These metrics were calculated only for functional groups containing ≥5 taxa and are not shown for groups with fewer than this. In all panels, functional groups are coloured according to broadly defined functional categories.

## Turnover in functional composition

Numerous functional traits are shared across multiple taxonomic groups, which should have led to occurrence patterns of functional groups that were largely robust to habitat degradation[24,25]. Yet instead, we found strong evidence of systemic changes to the functional composition of degraded tropical forest. Habitat degradation was associated with turnover from large to small taxa, specialist to generalist taxa, and terrestrial to arboreal taxa (Fig. 3a). We found no general pattern with respect to trophic level, with no evidence that predators were more susceptible to habitat degradation than herbivores. The impacts of habitat degradation were felt by functional groups that generate the full breadth of ecological processes in tropical forest ecosystems (Fig. 3).

All plant functional groups declined in occurrence as habitat degradation increased, with the most sensitive being those with low rates of photosynthetic activity measured in the field[29], including high-timber-value species in the Dipterocarpaceae. Pioneer tree species, including those with low wood density, might normally be expected to increase rather than decrease in occurrence in response to logging disturbance[38]. However, removal of very high amounts of biomass necessarily results in the extraction of a progressively higher proportion of standing trees[39], which inevitably includes species with low wood density. There was strong turnover in the body size of most animal taxa, with declines in the occurrence of large-bodied taxa such as the lowland litter frog *Leptobrachium abbotti* occurring across the entire degradation gradient, whereas the occurrence of small-bodied taxa such as the cyprinid fish *Barbonymus balleroides* began to increase in the more heavily degraded forest. Habitat generalists that exploit multiple strata within the forest (such as termites in the genus *Microcerotermes*), and dietary generalists that consume many types of prey (omnivores such as the bearded pig *Sus barbatus*), both increased in occurrence, whereas trophic specialists such as the rhinoceros hornbill *Buceros rhinoceros* declined in occurrence. Turnover in specific dietary types was highly variable. The occurrence of fruit and flower feeders declined as habitat degradation progressed, whereas seed and nectar feeders increased in occurrence. The occurrence of animals that feed on live wood and live roots also increased, whereas those that feed on dead plant material and fungus declined in occurrence. Finally, there was considerable turnover in the ability of taxa to exploit the various forest strata as forest degradation progressed (Fig. 3). The occurrence of arboreal birds and mammals, including the Bornean orangutan *Pongo pygmaeus*, increased along the first half of the degradation gradient, after which mammals that have below-ground prey such as the large

treeshrew *Tupaia tana* began to increase in occurrence. At the same time, the occurrence of terrestrial birds such as the argus pheasant *Argusianus argus* declined rapidly, followed by declines in the occurrence of aquatic invertebrates and mammals at high levels of forest degradation.

## Rules of thumb for conservation planning

We found that focusing on the conservation of either individual taxa or functional groups resulted in remarkably congruent locations for ecological thresholds, providing clear, empirically justified rules of thumb about exactly where to target conservation action. Together, our data indicate that actions designed to proactively avoid ecological change should be targeted at different points in the forest degradation gradient from those where reactive action should be used to reverse historic ecological change. Our data were collected from a single site, however, and taxon responses to habitat degradation can vary across geographical gradients[40,41] meaning the exact location of taxon-specific thresholds might similarly vary, so more studies of a similar nature will be required to strengthen confidence in the generality of our conclusions.

Forests that have lost less than 30% of their biomass retain very high biodiversity and ecological value, and can make an important contribution to the terrestrial and freshwater conservation estates. Proactive conservation decisions—actions designed to safeguard a habitat against further degradation—in these relatively lightly degraded forests could include adding them directly to the conservation estate by giving them protected area status[13], should that be a valid and equitable approach to conservation in the region[15]. Alternatively, depending on the local political and economic situation, maximum timber extraction rates could be set at levels that ensure the threshold is not passed, and might simultaneously consider protecting the three-dimensional structure of the forest, which also affects the biodiversity value of logged forests[16]. However, we stress that 30% biomass loss is not the same as 30% biomass extraction, as the former includes the collateral damage to a forest from logging activity that can be more than triple the extracted biomass of harvested timber alone[42]. Biomass extraction rates should then be set at targets considerably lower than 30%—perhaps as low as 10%—although the use of reduced-impact logging techniques might facilitate higher commercial extraction rates.

Forests that have lost between 30 and 68% of their biomass are likely to require a mix of conservation actions encompassing both proactive and reactive strategies, with reactive approaches increasing in importance as biomass loss progresses and ultimately passes the 68% threshold. The conservation gains that could be obtained from reactive conservation and forest restoration efforts—specific actions designed to reverse the degradation of a habitat—are likely to be highest where tree biomass has been reduced by more than two-thirds (68%). Assuming that the biodiversity and ecosystem functionality of a degraded forest will recover as forest biomass increases, then remedial actions such as underplanting, liana cutting and invasive species control are likely to have the greatest impact on occurrence patterns of both taxa and functional groups in these heavily degraded forests. Given such actions will accelerate the accumulation of carbon in degraded forests[43], funding for remedial actions might be raised through the sale of carbon credits[44].

There is no doubt, from our results and those of others[5], that primary forests are unique. Nonetheless, our data contribute to an emerging evidence base demonstrating that logged forests can and do retain high biodiversity[2,3,8] and ecological[1] value. Moreover, the ecological and biodiversity differences that do exist between primary forests and lightly logged forests can be small[5,11,45]. These results demand that we stop devaluing degraded tropical forests for what they have lost, and rather appreciate them for the many values they retain. The future of conservation across the tropics is highly dependent on human-modified habitats[4], and the way we choose to manage logged tropical forests will have a decisive role in stemming global biodiversity loss.

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

Robert M. Ewers[1✉], C. David L. Orme[1], William D. Pearse[1], Nursyamin Zulkifli[2], Genevieve Yvon-Durocher[3], Kalsum M. Yusah[4,5], Natalie Yoh[6,7], Darren C. J. Yeo[8,9], Anna Wong[10], Joseph Williamson[11,12], Clare L. Wilkinson[1,9], Fabienne Wiederkehr[1,13], Bruce L. Webber[14,15], Oliver R. Wearn[1,16], Leona Wai[17], Maisie Vollans[1,18], Joshua P. Twining[1,19], Edgar C. Turner[1,20], Joseph A. Tobias[1], Jack Thorley[20], Elizabeth M. Telford[21], Yit Arn Teh[22], Heok Hui Tan[8], Tom Swinfield[20], Martin Svátek[23], Matthew Struebig[6], Nigel Stork[24], Jani Sleutel[25], Eleanor M. Slade[26], Adam Sharp[1,27], Adi Shabrani[28,29], Sarab S. Sethi[1,30], Dave J. I. Seaman[6], Anati Sawang[31,32], Gabrielle Briana Roxby[1], J. Marcus Rowcliffe[33], Stephen J. Rossiter[11], Terhi Riutta[1,34,35], Homathevi Rahman[4], Lan Qie[1,36], Elizabeth Psomas[1,37], Aaron Prairie[1,38], Frederica Poznansky[1,39], Rajeev Pillay[40,41], Lorenzo Picinali[42], Annabel Pianzin[4], Marion Pfeifer[22], Jonathan M. Parrett[43], Ciar D. Noble[1,44], Reuben Nilus[45], Nazirah Mustaffa[4], Katherine E. Mullin[6], Simon Mitchell[6], Amelia R. Mckinlay[1], Sarah Maunsell[46], Radim Matula[47], Michael Massam[1,48], Stephanie Martin[1,49], Yadvinder Malhi[34], Noreen Majalap[45], Catherine S. Maclean[1], Emma Mackintosh[1,50], Sarah H. Luke[20,51,52], Owen T. Lewis[18], Harry L. Layfield[1,53], Isolde Lane-Shaw[1,54], Boon Hee Kueh[4], Pavel Kratina[11], Oliver Konopik[55], Roger Kitching[46], Lois Kinneen[46,56], Victoria A. Kemp[11], Palasiah Jotan[47], Nick Jones[57], Evyen W. Jebrail[4], Michal Hroneš[58], Sui Peng Heon[1,31], David R. Hemprich-Bennett[11,18], Jessica K. Haysom[6], Martina F. Harianja[20], Jane Hardwick[46,59], Nichar Gregory[1,60], Ryan Gray[31], Ross E. J. Gray[1], Natasha Granville[1], Richard Gill[1], Adam Fraser[1], William A. Foster[20], Hollie Folkard-Tapp[1], Robert J. Fletcher[40], Arman Hadi Fikri[4], Tom M. Fayle[11,61], Aisyah Faruk[62], Paul Eggleton[63], David P. Edwards[30,64], Rosie Drinkwater[11], Rory A. Dow[65,66], Timm F. Döbert[14,15,67], Raphael K. Didham[14,15], Katharine J. M. Dickinson[68], Nicolas J. Deere[6], Tijmen de Lorm[1], Mahadimenakbar M. Dawood[4], Charles W. Davison[1,69,70], Zoe G. Davies[6], Richard G. Davies[52], Martin Dančák[71], Jeremy Cusack[1,72], Elizabeth L. Clare[11,73], Arthur Chung[45], Vun Khen Chey[45], Philip M. Chapman[1,74], Lauren Cator[1], Daniel Carpenter[63], Chris Carbone[33], Kerry Calloway[63], Emma R. Bush[75], David F. R. P. Burslem[76], Keiron D. Brown[63], Stephen J. Brooks[63], Ella Brasington[1], Hayley Brant[1], Michael J. W. Boyle[1,77], Sabine Both[78], Joshua Blackman[11], Tom R. Bishop[1,79,80], Jake E. Bicknell[6], Henry Bernard[4], Saloni Basrur[6], Maxwell V. L. Barclay[63], Holly Barclay[81], Georgina Atton[82], Marc Ancrenaz[83,84], David C. Aldridge[20], Olivia Z. Daniel[1], Glen Reynolds[31] & Cristina Banks-Leite[1]

[1]Georgina Mace Centre for the Living Planet, Department of Life Sciences, Imperial College London, Ascot, UK. [2]Faculty of Forestry and Environment, Universiti Putra Malaysia, Seri Kembangan, Malaysia. [3]School of Physiology, Pharmacology and Neuroscience, University of Bristol, Bristol, UK. [4]Institute for Tropical Biology and Conservation, Universiti Malaysia Sabah, Kota Kinabalu, Malaysia. [5]Royal Botanic Gardens, Kew, Richmond, London, UK. [6]Durrell Institute of Conservation and Ecology (DICE), School of Anthropology and Conservation, University of Kent, Canterbury, UK. [7]The Nelson Institute for Environmental Studies, University of Wisconsin-Madison, Madison, WI, USA. [8]Lee Kong Chian Natural History Museum, National University of Singapore, Singapore, Singapore. [9]Department of Biological Sciences, National University of Singapore, Singapore, Singapore. [10]Malaysian Nature Society, Kuala Lumpur, Malaysia. [11]School of Biological and Behavioural Sciences, Queen Mary University of London, London, UK. [12]Centre for Biodiversity and Environment Research, Department of Genetics, Evolution and Environment, University College London, London, UK. [13]Institute of Microbiology, Department of Biology, ETH Zürich, Zurich, Switzerland. [14]School of Biological Sciences, The University of Western Australia, Crawley, Western Australia, Australia. [15]CSIRO Health and Biosecurity, Centre for Environment and Life Sciences, Floreat, Western Australia, Australia. [16]Fauna & Flora International, Hanoi, Vietnam. [17]Danau Girang Field Centre, Kinabatangan, Malaysia. [18]Department of Biology, University of Oxford, Oxford, UK. [19]New York Cooperative Fish and Wildlife Research Unit, Department of Natural Resources, Cornell University, Ithaca, NY, USA. [20]Department of Zoology, The David Attenborough Building, University of Cambridge, Cambridge, UK. [21]School of Geosciences, University of Edinburgh, Edinburgh, UK. [22]School of Natural and Environmental Sciences, Newcastle University, Newcastle upon Tyne, UK. [23]Department of Forest Botany, Dendrology and Geobiocoenology, Faculty of Forestry and Wood Technology, Mendel University in Brno, Brno, Czech Republic. [24]Centre for Planetary Health and Food Security, Griffith University, Brisbane, Queensland, Australia. [25]Department of Biology, Vrije Universiteit Brussel, Brussels, Belgium. [26]Asian School of the Environment, Nanyang Technological University, Singapore, Singapore. [27]Conservation & Fisheries Directorate, Ascension Island Government, Georgetown, St Helena Island. [28]Division of Biological Sciences, University of Montana, Missoula, MT, USA. [29]WWF-Malaysia, Kota Kinabalu, Malaysia. [30]Department of Plant Sciences, University of Cambridge, Cambridge, UK. [31]South East Asia Rainforest Research Partnership, Danum Valley Field Centre, Lahad Datu, Malaysia. [32]Sabah State Museum, Kota Kinabalu, Malaysia. [33]Institute of Zoology, Zoological Society of London, London, UK. [34]Environmental Change Institute, School of Geography and the Environment, University of Oxford, Oxford, UK. [35]Department of Geography, University of Exeter, Exeter, UK. [36]Department of Life Sciences, School of Life and Environmental Sciences, University of Lincoln, Lincoln, UK. [37]Oxitec, Abingdon, UK. [38]Department of Soil and Crop Sciences, Colorado State University, Fort Collins, CO, USA. [39]Centre for Ecology and Conservation, School of Biosciences, University of Exeter, Penryn, UK. [40]Department of Wildlife Ecology and Conservation, University of Florida, Gainesville, FL, USA. [41]Natural Resources and Environmental Studies Institute, University of Northern British Columbia, Prince George, British Columbia, Canada. [42]Dyson School of Design Engineering, Imperial College London, London, UK. [43]Faculty of Biology, Adam Mickiewicz University, Poznań, Poland. [44]School of Environmental Sciences, University of East Anglia, Norwich, UK. [45]Forest Research Centre, Sabah Forestry Department, Sandakan, Malaysia. [46]School of Environmental and Natural Sciences, Griffith University, Brisbane, Queensland, Australia. [47]Department of Forest Ecology, Faculty of Forestry and Wood Sciences, Czech University of Life Sciences Prague, Prague, Czech Republic. [48]School of Biosciences, The University of Sheffield, Sheffield, UK. [49]Field Programmes Department, Durrell Wildlife Conservation Trust, La Profonde Rue, Jersey. [50]Forest Research Institute, University of the Sunshine Coast, Sippy Downs, Queensland, Australia. [51]School of Biosciences, University of Nottingham, Loughborough, UK. [52]School of Biological Sciences, University of East Anglia, Norwich, UK. [53]School of Biological Sciences, University of Bristol, Bristol, UK. [54]Department of Wood and Forest Science, Laval University, Quebec, Quebec, Canada. [55]Department of Animal Ecology and Tropical Biology, Biocenter, University of Wuerzburg, Am Hubland, Würzburg, Germany. [56]Department of Sustainable Land Management, School of Agriculture, Policy and Development, University of Reading, Reading, UK. [57]Department of Mathematics, Imperial College London, London, UK. [58]Department of Botany, Faculty of Science, Palacký University, Olomouc, Czech Republic. [59]Marine Resources Unit, Department of Environment, Grand Cayman, Cayman Islands. [60]EcoHealth Alliance, New York, NY, USA. [61]Biology Centre of the Czech Academy of Sciences, Institute of Entomology, České Budějovice, Czech Republic. [62]Royal Botanic Gardens, Kew, Wakehurst, Haywards Heath, UK. [63]Department of Life Sciences, The Natural History Museum London, London, UK. [64]Ecology and Evolutionary Biology, School of Biosciences, University of Sheffield, Sheffield, UK. [65]Institute of Biodiversity and Environmental Conservation, Universiti Malaysia Sarawak, Kota Samarahan, Malaysia. [66]Naturalis Biodiversity Centre, Leiden, The Netherlands. [67]Faculty of Science, University of Alberta, Edmonton, Alberta, Canada. [68]Department of Botany, University of Otago, Dunedin, New Zealand. [69]Center for Biodiversity Dynamics in a Changing World (BIOCHANGE), Department of Biology, Aarhus University, Aarhus, Denmark. [70]Center for Ecological Dynamics in a Novel Biosphere (ECONOVO), Department of Biology, Aarhus University, Aarhus, Denmark. [71]Department of Ecology and Environmental Sciences, Faculty of Science, Palacký University, Olomouc, Czech Republic. [72]Okala, London, UK. [73]Department of Biology, York University, Toronto, Ontario, Canada. [74]BSG Ecology, Witney, UK. [75]Royal Botanic Gardens Edinburgh, Edinburgh, UK. [76]School of Biological Sciences, University of Aberdeen, Aberdeen, UK. [77]School of Biological Sciences, The University of Hong Kong, Hong Kong, Hong Kong. [78]School of Environmental and Rural Science, Faculty of Science, Agriculture, Business and Law, University of New England, Armidale, New South Wales, Australia. [79]Department of Zoology and Entomology, University of Pretoria, Pretoria, South Africa. [80]School of Biosciences, Cardiff University, Cardiff, UK. [81]School of Science, Monash University, Subang Jaya, Malaysia. [82]Faculty of Health Sciences, University of Bristol, Bristol, UK. [83]Borneo Futures, Bandar Seri Begawan, Brunei. [84]Kinabatangan Orang-Utan Conservation Programme, Kota Kinabalu, Malaysia. ✉e-mail: r.ewers@imperial.ac.uk

## Methods

All data manipulation, data analysis and construction of figures were conducted in the R v.4.02 computing environment[46], using the packages ape (v.5.0)[47], betareg (v.3.1-4)[48], dplyr (v.1.1.4)[49], lme4 (v.1.1-35.1)[50], lmtest (v.0.9-40)[51], lubridate (v.1.9.3)[52], MASS (v.7.3-60.0.1)[53], openxlsx (v.4.2.5.2)[54], paletteer (v.1.6.0)[55], pastecs (v.1.4.2)[56], png (v.0.1-8)[57], raster (v.3.6-26)[58], reshape2 (v.1.4.4)[59], rgdal (v.1.6-7)[60], rgeos (v.0.6-4)[61], safedata (v.1.1.3)[62], scales (v.1.3.0)[63], sf (v.1.0-15)[64], spgwr (v.0.6-36)[65], stringr (v.1.5.1)[66] and strucchange (v.1.5-3)[67].

### Taxa records and functional groups

We summarized taxon responses from 8,130 combinations of surveys and taxa. We compiled biodiversity data from 55 published data sources[68–122] (Supplementary Table 1), from which we extracted presence–absence data following the methods of ref. 123. Previous analyses of multi-taxa biodiversity data have demonstrated that comparisons of presence–absence data among taxa are more robust than analyses of abundance data[23,124]. Moreover, abundance data were not available for all taxa, meaning that presence–absence data are the highest-level data that allowed us to use exactly the same analysis method for all taxa. Data sources that sampled multiple years were split into separate, annual surveys, allowing us to more accurately align biodiversity observations with forest degradation measurements taken at different time points, and to account for year-to-year variation in taxon-specific responses to the same ecological gradient[123]. Data sources that included multiple sampling methods were also split into separate, method-specific surveys[123]. This process resulted in a total of 127 surveys being used for analysis.

Not all taxa in all surveys were identified to species or morphospecies level. We retained data on taxa identified to higher taxonomic levels because these could often be confidently placed into valid functional groups for analysis. Our data encompassed 4,691 taxa distributed widely across the terrestrial tree of life (Extended Data Fig. 1), of which 1,777 were identified to species and a further 2,288 to morphospecies. We restricted our statistical analyses to 1,681 taxa that had ≥5 occurrences (Extended Data Fig. 1), of which more than half (n = 946) were observed in more than 1 survey (Extended Data Fig. 2). Sensitivity analyses on these same data have demonstrated that a cutoff of five occurrences is appropriate to generate consistently reliable results[123].

**Taxonomic and functional groups.** We aggregated taxa into high-level taxonomic and functional groups to examine group-specific trends. First, we categorized taxa into ten taxonomic groups for separate analysis (plants, arachnids, non-ant insects, ants, other invertebrates, mammals, birds, reptiles, amphibians and fish). Second, we compiled information on directly recorded morphological, functional and physiological traits for as many taxa as we could, which we used to allocate taxa into 126 functional groups (Supplementary Table 2). In doing so, we relied heavily on previously published surveys[68,69,125–130], literature reviews and expert knowledge.

We included International Union for Conservation of Nature Red List status[131], which we collapsed into two categories: threatened (Critically Endangered, Endangered or Vulnerable) or not threatened (Least Concern, Lower Risk or Near Threatened). For plant taxa, we obtained data on wood density and photosynthesis rates[68], and used data on leaf area, leaf dry matter content and specific leaf area to estimate the strength of their association with each of three life history strategies: competitor, stress-tolerator or ruderal[132]. All plant traits were continuous, which we categorized into two groupings for analysis (low and high according to whether trait values were below or above the median respectively).

For animal taxa, we compiled data on body mass for mammals[125], birds[133], fish[126] and beetles[127] from previously published surveys, estimated amphibian body mass from snout–vent length measurements[134], and estimated ant body mass using a combination of morphometric data[59] and published scaling relationships[66]. Body mass was categorized into three groupings (low, medium and high) separately for each taxonomic group. Grouping boundaries were set by $\log_{10}$-transforming body mass and dividing taxa into three equal quantiles.

Animal taxa were assigned categories for physiology (endotherm or ectotherm), development (direct or indirect), sociality (solitary, pair, social or eusocial) and movement mode (winged, legged or legless). We used published records and expert knowledge to record non-mutually exclusive categories of forest strata use, classified as the strata where that taxon forages for food (subterranean, ground-dwelling, understorey, canopy-dwelling or aquatic), trophic level (saprophage, producer, herbivore, carnivore, parasite, parasitoid) and 21 diet categories (soil feeder, coprophage, necrophage, detritivore, saprophage, algivore, mycophage, rhizophage, folivore, florivore, nectarivore, palynivore, frugivore, granivore, xylophage, phloeophage, bacteriophage, invertivore, vertivore, piscivore, haematophage). For each of these last three functional traits, we counted the number of categories associated with each taxon, and categorized taxa as having either low or high generalism according to whether they fell above or below the median value for that trait.

### Quantifying forest degradation

We followed the protocols described previously[123] to develop a quantitative metric of forest degradation. In brief, data were collected at the Stability of Altered Forest Ecosystems Project[26] study site in Sabah, Malaysia. Taxa were sampled at sites that varied in the extent of historical disturbance from unlogged, old-growth forest through to salvage-logged forest and into deforested sites converted into oil palm plantations. We based our degradation metric on above-ground carbon density (ACD; Mg ha$^{-1}$) derived from airborne LiDAR data[135,136]. ACD values varied between 1 Mg ha$^{-1}$ in cleared areas to a maximum of 273 Mg ha$^{-1}$ in unlogged forest. For ease of interpretation, we converted ACD into a metric representing the percentage reduction in biomass relative to unlogged forest. We set the value of unlogged forest (0% biomass removal) to be the median biomass density observed in unlogged forest (230 Mg ha$^{-1}$). We chose to report values as a percentage as opposed to megagrams per hectare as it is more easily transferable to other tropical forests where the maximum ACD may vary[137]. Forest degradation was quantified at two time points that approximately bracketed a salvage logging operation in the project area—November 2014[135] and April 2016[136]—and taxa were analysed using the forest degradation values that were most closely matched in time to the date of the survey in which the taxon was observed.

The occurrence of a taxon at a given site is almost certainly a response to habitat conditions in a wider radius surrounding that site, so we calculated a spatial average to use as our predictor variable in analyses[33]. We selected all pixels within a radius of 250, 500, 1,000, 2,000 and 4,000 m, respectively, around each sample site. Pixels within the buffer area were averaged, with pixels weighted using a Gaussian distance weighting to ensure that those located close to the sample site carried more weight than those located further away. The Gaussian distance weighting ($W_g$) was given by the equation:

$$W_g = e^{\left(-\left(\frac{d}{h}\right)\right)^2}$$

in which $d$ represents distance from the central sample site and $h$ gives the bandwidth that was calculated as the maximum buffer distance divided by 100 (ref. 138).

### Quantifying and summarizing responses

We focus our analyses on the response patterns of individual taxa or functional groups and not aggregated metrics such as species richness[139] or coarse, vote-counting comparisons of the number of positively versus negatively affected taxa[140]. We take this approach

because turnover in the identity of taxa and functions are more sensitive measures of changes in biodiversity and ecosystem function. We focus instead on the locations of significant changes in response along the forest degradation gradient as opposed to the signs of those changes.

We test for two conservation-relevant patterns of change in the responses of individual taxa to forest degradation (Extended Data Fig. 3)−change points: the point at which forest degradation first exerts a discernible impact on the occurrence pattern of a taxon or functional group[141]; and maximum rate points: the point along the degradation gradient where the rate of change in occurrence is most rapid.

We use the aggregation of change points across taxa and functional groups along the forest degradation gradient to identify thresholds for prioritizing proactive conservation, whereas the aggregation of maximum rate points indicates locations where relatively small changes in habitat quality can have the largest impact on the system. If the pattern by which biodiversity recovers from logging is the reverse of the pattern by which it is affected by logging−that is, if there is no hysteresis[37]−then maximum rate points represent thresholds at which reactive conservation actions, such as forest restoration, are likely to be most effective. This is because conservation actions that add small amounts of biomass to the forest are expected to result in the largest collective change in the occurrence patterns of the affected taxa.

**Occurrence models.** We standardized all taxon observations to presence–absence data. To generate equivalent data for functional groups, within each survey we aggregated the presences of all taxa that belonged to a particular functional group. For each taxon and functional group, we then determined which survey(s) contained relevant data and combined all observations into a single data frame for analysis. Only taxa or functional groups that had ≥5 occurrence records were analysed, and this is the threshold value that results in repeated single-year surveys having the most consistent ecological results[123]. All individual taxa and functional groups were analysed independently of each other.

All models tested for an effect of percentage forest degradation on the probability of occurrence. Forest degradation was calculated at each of the five buffer sizes, and we selected the most appropriate spatial scale using the Akaike information criterion[53]. Statistical significance of the best model was determined with a log-likelihood ratio test comparing the best model to a null model. We tested for a main, linear effect of forest degradation alone. This was because visual inspection and diagnostic plots of exploratory analyses containing a polynomial term failed to identify clear cases of taxa that had peaks in occurrence at intermediate levels of biomass removal.

If a given taxon or functional group was present in more than one survey, we first used a binomial generalized linear mixed model (GLMM) including a random intercept term for survey identity. If GLMMs failed to converge, or if the taxon or functional group was present in only a single survey, we used binomial generalized linear models (GLMs). We were able to fit GLMMs to 798 out of the 946 taxa that were observed in multiple studies (84% of fitted models) and 72 functional groups (59%). The main reason by which GLMMs failed to converge was because taxa or functional groups observed in multiple datasets were not necessarily observed equally in all datasets, and low numbers of observations in one or more surveys can limit the ability of a GLMM to estimate survey-specific random effects.

We opted not to use modelling methods that directly control for detectability, as such models routinely failed to converge in preliminary analyses. This problem is often encountered for analyses of tropical biodiversity in which many species are rare and have low detection probabilities[142]. We note, however, that detectability models of species occupancy patterns along ecological gradients do not differ greatly from models that ignore detection probability[142], so we do not expect our choice of approach will notably influence our key results.

**Maximum rate points.** We used the first derivatives of fitted models to find the point along the forest degradation gradient where the predicted rate of change in occurrence is most rapid[141], which we termed the 'maximum rate point' (Extended Data Fig. 3c). This point was numerically estimated by identifying the point at which the predicted occurrence pattern from the binomial GLM had the highest absolute slope (as represented by the root of the second derivative), and corresponds to the point along the habitat degradation gradient where the probability of occurrence is 50%. We used absolute slope as occurrence patterns may either increase or decrease along the forest degradation gradient, resulting in positive or negative slopes, respectively.

**Change points.** We used the second derivatives of fitted models to find change points of the fitted binomial models (Extended Data Fig. 3c,d), which represent the point along the forest degradation gradient where the rate of change in occurrence is itself changing the fastest[141]. As with the maximum rate points, these were numerically estimated by identifying the point at which the first derivative of the binomial GLM had the highest absolute slope (as represented by the root of the third derivative).

Binomial GLMs with significant slopes have a change point on either side of the maximum rate point, and we focused our analyses on the point at the higher value of forest quality (lowest amount of biomass reduction in Extended Data Fig. 3). These represent the change points at which taxa first begin to respond to reductions in forest quality. Change points are undefined for models with no significant slope.

**Taxonomic bias in results.** Although the taxa we examined were diverse and are widely distributed across the tree of life (Extended Data Fig. 1), they are not evenly distributed across the tree of life. If the different taxa exhibit consistent variation in the pattern of their responses, this taxonomic bias might affect our overall conclusions. To test for this, we modelled both maximum rate points and change points as a function of taxonomic group, and used log-likelihood ratio tests to compare both models against a null model. There was no significant effect in either case (change points: $\chi^2_9 = 2.79$, $P = 0.97$; maximum rate points: $\chi^2_9 = 9.78$, $P = 0.37$), indicating that taxonomic bias in our dataset is unlikely to influence the interpretation of our results.

**Temporal bias in results.** Environmental conditions might influence the outcome of ecological studies[143]. If the surveys we analyse here are unequally distributed through time, and taxon responses to habitat degradation are time-dependent, then temporal autocorrelation might influence our conclusions. In a separate analysis of the same data used in this study, we have quantified this effect and demonstrated that it is not a concern[123]. We examined whether taxon-specific occurrence patterns across the habitat degradation gradient varied among surveys and years. We found that although occurrence patterns do vary among surveys, there was no consistent signal of survey year on those patterns. Specifically, the number of years between two surveys had no significant impact on the probability of two surveys reporting statistically indistinguishable response patterns.

Long-term shifts in the composition of forest communities might mean that the biodiversity patterns we associate with primary forest in our data are themselves depauperate relative to historical patterns[144]. Similarly, the complex logging history of our study site with repeated, but unequally distributed, rounds of logging means that many sites have been through multiple stages of degradation separated by partial recovery[145,146]. Our data are not sufficient to quantify historical patterns of occupancy or the impact of time lags on trajectories of occupancy, so we are unable to directly test for these effects. Nonetheless, long-term declines and local extinction of megafauna such as the Sumatran rhino *Dicerorhinus sumatrensis harrissoni*[147] make it likely that a shifting baseline is a valid concern at our study site. However, we have no way of knowing whether the rates of biodiversity change

from the processes that might generate baseline shifts will be the same or different in primary and logged forest. Consequently, we can only emphasize that our analyses are based on a space-for-time substitution, which makes the implicit assumption that the effects of habitat degradation we quantify are additional to, and do not interact with, any other processes contributing to long-term biodiversity change.

**Identifying thresholds.** We fitted density curves to model the distribution of taxa and functional group change points along the forest degradation gradient. Density curves were fitted using the kernel density estimation function with default settings in the 'stats' package[46]. Estimates were extracted, and we used breakpoint regression on the fitted density distributions to identify the number and location of thresholds in aggregated biodiversity and functional group responses to forest degradation.

Thresholds differ from the analysis of individual change points in that they are based on the aggregation of all change points. Whereas change point analysis identified locations where the occurrence pattern of an individual taxon changes, the thresholds identified here represent locations where there is a change in the accumulated responses of the 1,681 taxa or 126 functional groups. Two classes of thresholds are possible: breakpoints signalling either an increase or decrease in the rate of accumulation of affected taxa or functional groups. The former are acceleration points that signify locations at which the situation becomes worse, in that the rate at which the number of affected taxa or functional groups begins to increase (or the rate of decline begins to slow down) as forest degradation increases.

We repeated this approach using the distribution of maximum rate points for both taxa and functional groups. In all cases, the breakpoint regression identified an optimal model containing two acceleration breakpoints. We set the threshold for proactive and reactive conservation to be based on the first and last acceleration points, respectively. For each type of conservation, there were four proactive and four reactive thresholds estimated; one each for taxa change points, taxa maximum rate points, functional group change points and functional group maximum rate points. To obtain an aggregate threshold for proactive and reactive conservation, we used the mean of these four values.

We used bootstrapping to estimate a 95% confidence interval around these means by resampling the fitted models 100 times and estimating the 2.5 and 97.5% quantiles around the threshold estimates.

**Vulnerability of taxonomic and functional categories to forest degradation.** We combined two metrics to estimate the relative vulnerability of taxonomic and functional groups to forest degradation: probability of impact, defined as the proportion of taxa within that group that exhibited a change point; and severity of impact, defined as the mean location of change points among taxa within that group. Specifically, probability of impact (PI) is calculated as:

$$PI = \frac{\sum_{t=1}^{N} I_t}{N}$$

in which $N$ represents the number of taxa within that taxonomic category, $I_t$ is a binary outcome representing whether taxon $t$ is significantly affected by forest degradation, calculated as:

$$I_t = \begin{cases} 1, \text{ if } p_t < 0.05 \\ 0, \text{ if } p_t \geq 0.05 \end{cases}$$

and $p_t$ is the $P$ value from the analysis of taxon $t$'s occurrence pattern in response to forest degradation. Taxonomic categories with large numbers of affected taxa have high probability of impact values. Correlation analyses demonstrated that there was no impact of sample size (the number of taxa per group) on probability of impact for either

taxonomic groups ($r = -0.21$, d.f. = 8, $P = 0.56$) or functional groups ($r = -0.11$, d.f. = 45, $P = 0.45$).

Severity of impact (SI) is calculated as:

$$SI = 1 - \frac{\sum_{t=1}^{N} CP_t}{N \times 100}$$

in which $CP_t$ is the change point of taxon $t$'s response pattern to forest degradation (Extended Data Fig. 3c), and scales such that categories containing many taxa that tend to be affected after the removal of small amounts of biomass have high severity of impact values. The change point for taxa that are not affected by forest degradation ($p_t > 0.05$) is undefined, but excluding them from the severity of impact calculation would skew severity estimates: categories with large numbers of unaffected taxa would retain the severity value calculated from the small number of affected taxa. We therefore assigned unaffected taxa a change point of 100 before calculating severity. This value indicates that the taxon is not affected until 100% of biomass has been removed, and represents the least-sensitive, real-world change point value.

Both probability of impact and severity of impact are bounded at zero and one, and we combined them into a single metric of vulnerability ($V$) calculated as

$$V = PI \times SI$$

which is also bounded at zero and one. Taxonomic categories containing a high proportion of taxa that are affected by low amounts of biodiversity loss have high vulnerability values. By contrast, categories in which a low proportion of taxa are affected, and the taxa that are affected experience change points only after the removal of large amounts of forest biomass, have the lowest vulnerability values.

To summarize functional vulnerability, we categorized functional groups into ten higher-level categories: Red List status, habitat strata, physiology, development, sociality, movement, diet, trophic, body mass and plant traits (for all plant-specific functional groups). Within each category, we treated the individual functional groups as replicates, allowing us to calculate the probability of impact, severity of impact and vulnerability of broadly categorized functional responses.

**Inclusion and ethics.** All data used were collected in Malaysia. Non-Malaysian researchers conducting field work collaborated with local researchers throughout the research process. All local collaborators were invited to co-author this publication, as were all Malaysian research students involved in data collection.

### Reporting summary

Further information on research design is available in the Nature Portfolio Reporting Summary linked to this article.

## Data availability

Datasets used in these analyses were published separately[68–122,135,136]; information regarding individual Zenodo repositories is included in Supplementary Table 1.

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

**Acknowledgements** This study was supported by funding to the Stability of Altered Forest Ecosystems Project by the Sime Darby Foundation. Research permission and site access were provided by the Maliau Basin Management Committee, the Sabah Foundation, Benta Wawasan, Sabah Softwoods, the Innoprise Foundation, the Sabah Forestry Department and the Sabah Biodiversity Centre. R.M.E. is supported by the NOMIS Foundation. Data collection was financed by Australian Research Council grant DP140101541; Bat Conservation International; the British Council Newton-Ungku Omar Fund 216433953; British Ecological Society grant 3256/4035; the Cambridge Trust; the Cambridge University Commonwealth Fund; the Czech Science Foundation (14-32302S); the European Research Council (281986); the European Social Fund and the Czech Republic (CZ.1.07/2.3.00/20.0064); the Fundamental Research Grant Scheme (FRG0302-STWN-1/ 2011), Ministry of Higher Education, Malaysia; FFWS CZU (IGA number A_26_22); the Jardine Foundation; Malaysia Industry Group for High Technology (216433953); the Ministry of Education, Youth and Sports of the Czech Republic (INTER-TRANSFER LTT19018); the Panton Trust; the Primate Society of Great Britain; ProForest; Royal Society of London grant RG130793; the Sime Darby Foundation; the S.T. Lee Fund; the Sir Philip Reckitt Educational Trust; the Tim Whitmore Fund; the Universiti Malaysia Sabah; the University of East Anglia; the University of Kent; the University of Florida Institute of Food and Agricultural Sciences; UK Research and Innovation Natural Environment Research Council grants NE/H011307/1, NE/K016253/1, NE/K016407/1, NE/K016148/1, NE/K0106261/1, NE/K015377/1, NE/L002515/1, NE/L002582/1 and NE/P00363X/1 and studentship 1122589; the Varley Gradwell Travelling Fellowship; and the World Wildlife Fund for Nature. Data collection was supported by R. Adzhar, A. Afendy, N. Arumugam, S. Benedick, V. Bignet, S. Butler, K. Graves, H. E. Hah, H. Heroin, A. Kendall, H. H. Mahsol, D. Mann, J. Miller, S. Milne, J. Mumford, D. Norman, H. Rossleykho, D. Shapiro, K. Sieving, J. Sugau, B. Udell, B. E. Yahya and M. A. Zakaria.

**Author contributions** R.M.E. designed the study, conducted the analyses and drafted the manuscript. C.D.L.O., W.D.P., G.R. and C.B.-L. supported the data analysis, helped interpret the results and edited the manuscript. All other authors contributed field data and checked the manuscript.

**Competing interests** The authors declare no competing interests.

**Additional information**
**Correspondence and requests for materials** should be addressed to Robert M. Ewers.

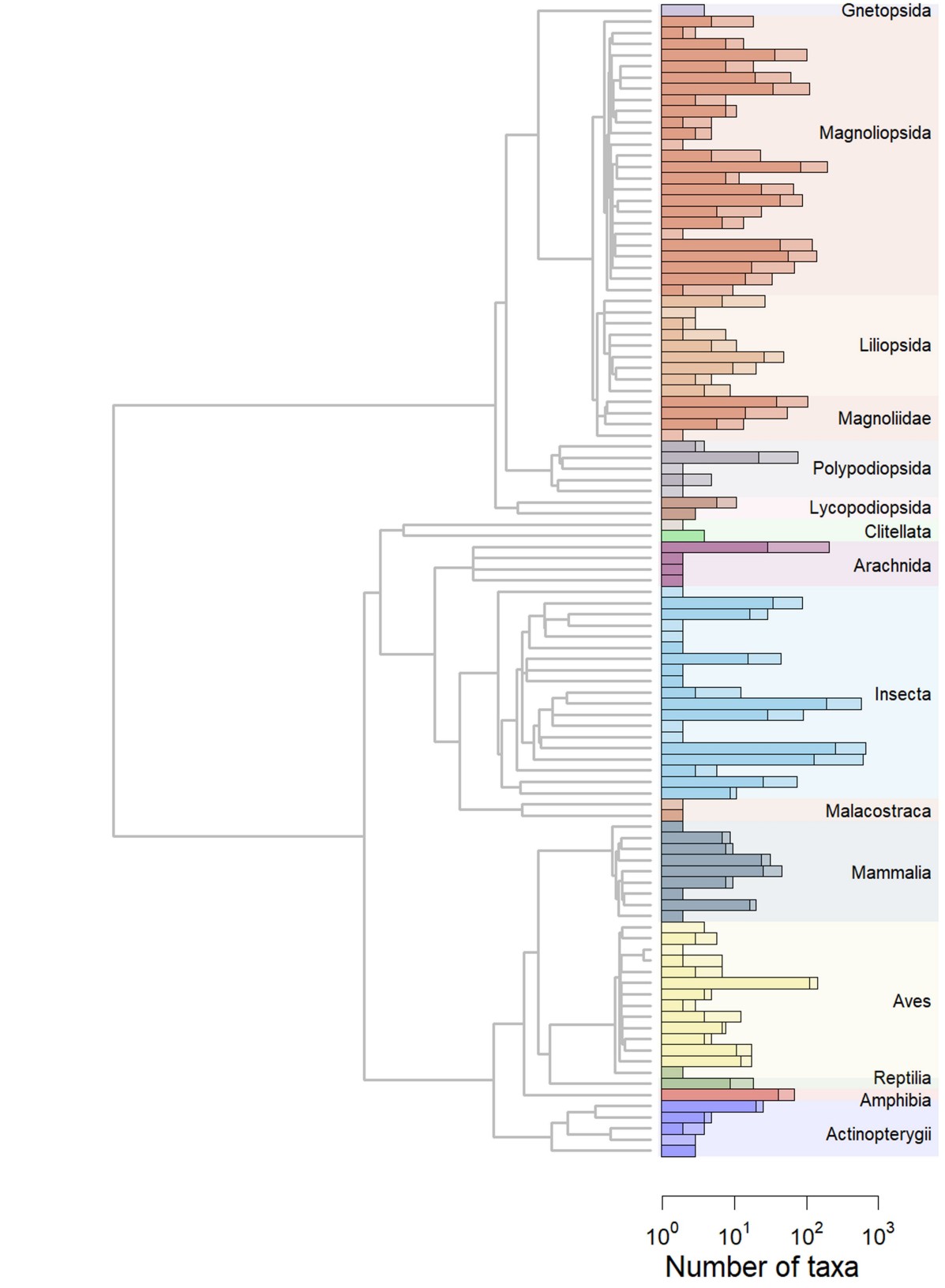

**Extended Data Fig. 1 | Phylogenetic super-tree[54] showing the 103 orders represented in the full set of biodiversity surveys.** Of the 103 orders, 86 had at least one taxon with enough occurrence observations to be analysed. Bar length represents the number of taxa per order (light shading), and the number of taxa that were analysed (dark shading). Bars are presented on a $\log_{10}$-scale and are coloured according to taxonomic class.

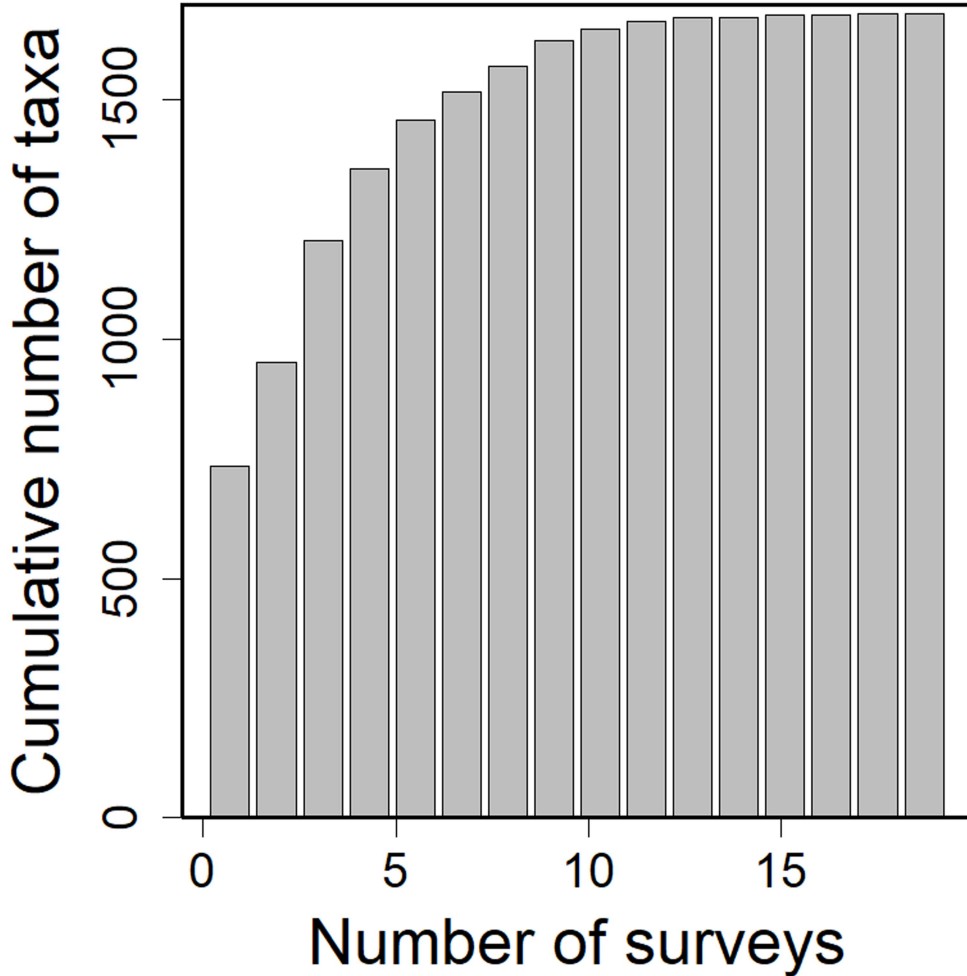

**Extended Data Fig. 2 | Distribution of number of surveys per taxon for the 1,681 modelled taxa.** Of the taxa, 731 (44 %) were represented in a single survey, and the remaining 946 (56 %) were represented in multiple surveys.

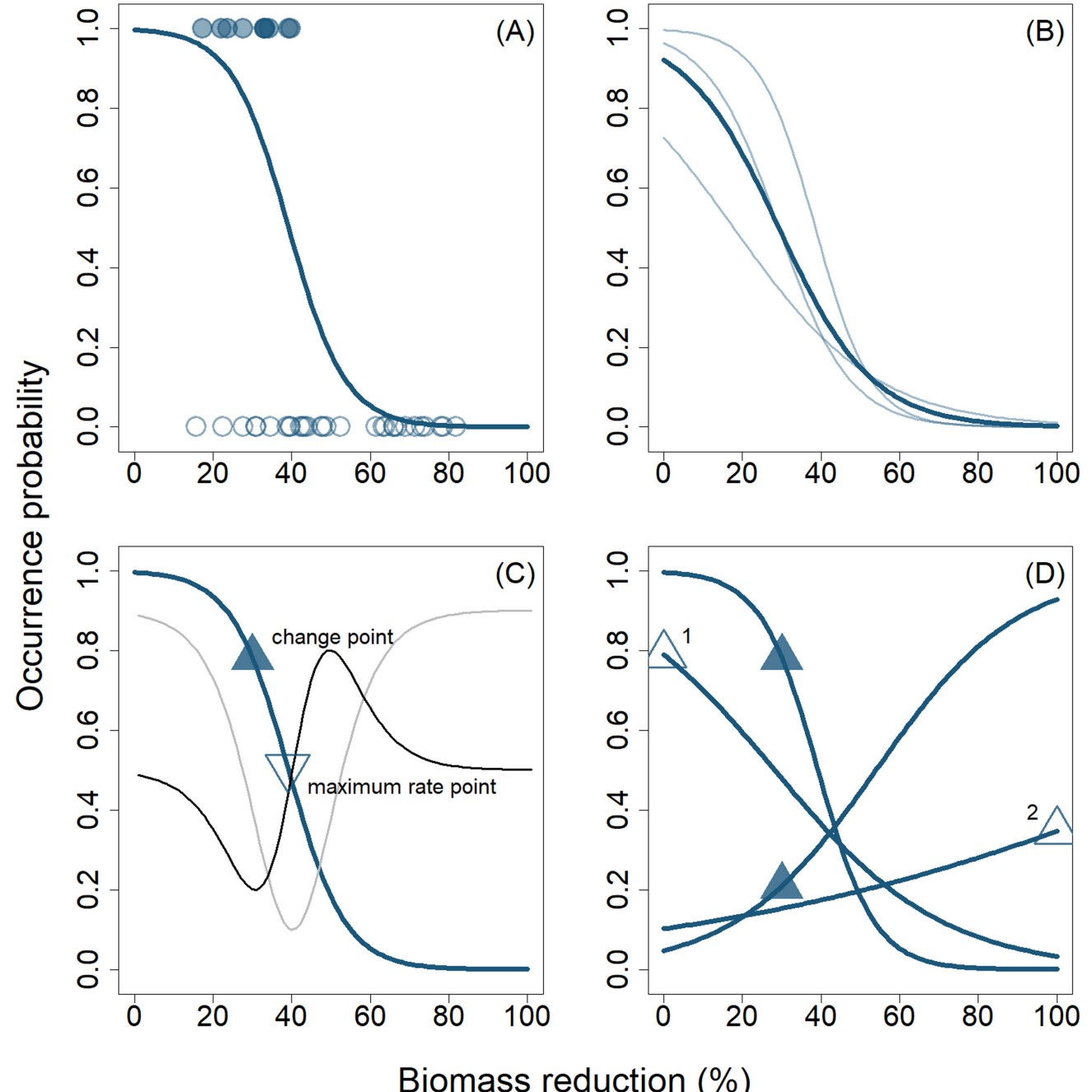

**Extended Data Fig. 3 | Visualisation of the data analysis process. (A)** For a given taxon in a given survey, we modelled taxon occurrence using presence (filled circles) and absence (open circles) data collected from individual surveys. Fitted occurrence probabilities were predicted across the forest degradation gradient. Forest degradation is represented as a percentage reduction in aboveground biomass, where zero represents the median biomass in unlogged forest. **(B)** Some taxa were observed in multiple surveys (represented by semi-transparent lines, here fitted as survey-specific linear models), each of which could have a different occurrence pattern[123]. In these cases, we used a mixed effect model to combine observations across all datasets, generating a single model of that taxon's occurrence pattern that was used to determine turning and maximum rate points (thick line). **(C)** The second derivative (black line; y-axis values not shown) of the fitted curve (thick blue line) was used to detect change points (filled triangle), which signify the point at which forest degradation first exerts a discernible impact on taxon occurrence[141]. Similarly, the first derivative (grey line; y-axis values not shown) was used to detect the point along the forest degradation gradient where the rate of change in occurrence of that taxon was the greatest (open triangle). **(D)** The approach used in panel (C) was applied to all taxa and functional groups. Two rules were used to record change points that fell outside of the survey's forest degradation range (open triangles): if the change point occurred below or above the range of feasible values it was truncated to 0% or 100% respectively (labelled 1 and 2 on the figure).

| | |
|---|---|

# Reporting Summary

## Statistics

For all statistical analyses, confirm that the following items are present in the figure legend, table legend, main text, or Methods section.

| n/a | Confirmed | |
|---|---|---|
| ☐ | ☒ | The exact sample size (*n*) for each experimental group/condition, given as a discrete number and unit of measurement |
| ☐ | ☒ | A statement on whether measurements were taken from distinct samples or whether the same sample was measured repeatedly |
| ☐ | ☒ | The statistical test(s) used AND whether they are one- or two-sided *Only common tests should be described solely by name; describe more complex techniques in the Methods section.* |
| ☐ | ☒ | A description of all covariates tested |
| ☐ | ☒ | A description of any assumptions or corrections, such as tests of normality and adjustment for multiple comparisons |
| ☐ | ☒ | A full description of the statistical parameters including central tendency (e.g. means) or other basic estimates (e.g. regression coefficient) AND variation (e.g. standard deviation) or associated estimates of uncertainty (e.g. confidence intervals) |
| ☐ | ☒ | For null hypothesis testing, the test statistic (e.g. *F*, *t*, *r*) with confidence intervals, effect sizes, degrees of freedom and *P* value noted *Give P values as exact values whenever suitable.* |
| ☒ | ☐ | For Bayesian analysis, information on the choice of priors and Markov chain Monte Carlo settings |
| ☒ | ☐ | For hierarchical and complex designs, identification of the appropriate level for tests and full reporting of outcomes |
| ☐ | ☒ | Estimates of effect sizes (e.g. Cohen's *d*, Pearson's *r*), indicating how they were calculated |

*Our web collection on statistics for biologists contains articles on many of the points above.*

## Software and code

Policy information about availability of computer code

| Data collection | Data were compiled from online databases using the safedata R package (v 1.1.3) |
|---|---|
| Data analysis | All data manipulation, data analysis and construction of figures were conducted in the R v4.02 computing environment, using the packages ape v5.0, betareg v3.1-4, dplyr v1.1.4, lme4 v1.1-35.1, lmtest v0.9-40, lubridate v1.9.3, MASS v7.3-60.0.1, openxlsx v4.2.5.2, paletteer v1.6.0, pastecs v1.4.2, png v0.1-8, raster v3.6-26, reshape2 v1.4.4, rgdal v1.6-7, rgeos v0.6-4, safedata v1.1.3, scales v1.3.0, sf v1.0-15, spgwr v0.6-36, stringr v1.5.1 and strucchange v1.5-3. |

For manuscripts utilizing custom algorithms or software that are central to the research but not yet described in published literature, software must be made available to editors and reviewers. We strongly encourage code deposition in a community repository (e.g. GitHub). See the Nature Portfolio guidelines for submitting code & software for further information.

## Data

Policy information about availability of data

All manuscripts must include a data availability statement. This statement should provide the following information, where applicable:

- Accession codes, unique identifiers, or web links for publicly available datasets
- A description of any restrictions on data availability
- For clinical datasets or third party data, please ensure that the statement adheres to our policy

| Datasets used in these analyses were published separately. Citations are provided in Table S1. |
|---|

# Research involving human participants, their data, or biological material

Policy information about studies with human participants or human data. See also policy information about sex, gender (identity/presentation), and sexual orientation and race, ethnicity and racism.

| | |
|---|---|
| Reporting on sex and gender | NA |
| Reporting on race, ethnicity, or other socially relevant groupings | NA |
| Population characteristics | NA |
| Recruitment | NA |
| Ethics oversight | NA |

Note that full information on the approval of the study protocol must also be provided in the manuscript.

# Field-specific reporting

Please select the one below that is the best fit for your research. If you are not sure, read the appropriate sections before making your selection.

☐ Life sciences ☐ Behavioural & social sciences ☒ Ecological, evolutionary & environmental sciences

For a reference copy of the document with all sections, see nature.com/documents/nr-reporting-summary-flat.pdf

# Ecological, evolutionary & environmental sciences study design

All studies must disclose on these points even when the disclosure is negative.

| | |
|---|---|
| Study description | This is a meta-analysis that combines data from 127 separate field surveys, each of which examined the distribution of organisms along a gradient of forest degradation. |
| Research sample | The sample is 127 individual datasets, each of which had a bespoke number of sample sites and replicates. |
| Sampling strategy | We sampled all datasets published on the SAFE Project Zenodo community |
| Data collection | Datasets used in this meta-analysis were compiled from online sources. Original field data were collected using a wide variety of methods according to the focus of the particular studies. All authors of the original data are included as authors on the manuscript. |
| Timing and spatial scale | All original datasets used in this meta-analysis were collected between 2010-2020 from the SAFE Project study site in Malaysia, which has a spatial extent of approximately 10,000 ha. |
| Data exclusions | No data were excluded |
| Reproducibility | Many of the individual taxa we analyse were detected in more than one field survey, meaning our analyses represent taxon-level responses that are averaged across multiple surveys. |
| Randomization | Datasets were grouped according to whether or not they had shared taxa in common. No other groupings of data were used. |
| Blinding | NA |

Did the study involve field work? ☒ Yes ☐ No

# Field work, collection and transport

| | |
|---|---|
| Field conditions | Average annual rainfall at the site is ~3060 mm, and average annual temperature is ~23 degrees Celcius. |
| Location | The study site is located at roughly 116 degrees East 4 degrees North. The average altitude of sampling points is ~400 masl. |
| Access & import/export | Original datasets used in this meta-analysis were collected buy authors in line with Malaysian requirements for research in the state of Sabah. This included working productively with local collaborators, obtaining permissions from land owners, and obtaining research permission from the Sabah Biodiversity Centre. |

| Disturbance | Original studies used in this meta-analysis were collected using a wide variety of non-invasive and invasive survey techniques. Each was appropriate and proportional to the data being collected, and none generated lasting disturbance to the ecosystem. |
|---|---|

# Reporting for specific materials, systems and methods

We require information from authors about some types of materials, experimental systems and methods used in many studies. Here, indicate whether each material, system or method listed is relevant to your study. If you are not sure if a list item applies to your research, read the appropriate section before selecting a response.

## Materials & experimental systems

| n/a | Involved in the study |
|---|---|
| ☒ ☐ | Antibodies |
| ☒ ☐ | Eukaryotic cell lines |
| ☒ ☐ | Palaeontology and archaeology |
| ☒ ☐ | Animals and other organisms |
| ☒ ☐ | Clinical data |
| ☒ ☐ | Dual use research of concern |
| ☒ ☐ | Plants |

## Methods

| n/a | Involved in the study |
|---|---|
| ☒ ☐ | ChIP-seq |
| ☒ ☐ | Flow cytometry |
| ☒ ☐ | MRI-based neuroimaging |

