## [Peer Review File · Nature]

Manuscript Title: Ecological thresholds for adding degraded tropical rainforests to the conservation estate

Reviewer Comments & Author Rebuttals

Reviewer Reports on the Initial Version:

Referees' comments:

Referee #1 (Remarks to the Author):

A. Summary of the key results

This study presents an interesting approach to the multi-dimensional effects of tropical forest degradation (and in particular selective logging) on forest composition, at both taxonomic and functional levels. To do so, it analyses how species occurrence patterns change within a gradient of forest degradation. The authors provide thresholds of biomass removal above below which degraded forests could significantly contribute to biodiversity conservation, and above which logged forests would need restoration actions to maintain high diversity levels. The conclusions should be of interest not only to ecologists, but also to policy makers and forest managers seeking to establish general rules for biodiversity conservation in anthropised forest landscapes.

B. Originality and significance: if not novel, please include reference

The originality of this study lies in the use of a large number of taxa across a wide range of life forms, from plants to insects, mammals, etc., which to my knowledge has never been done before.

C. Data & methodology: validity of approach, quality of data, quality of presentation

a) As defined, the probability of impact could depend on the sampling effort (e.g. the number of observations) in different taxonomic and functional groups. Is there a link between these two variables?

b) I suggest using the median rather than the mean to estimate impact severity, unless there is a good reason to use the mean. The median is less sensitive to outliers, and the effect of outliers could explain some discrepancies between impact severity and taxa change points in Fig. 3.

c) By substituting the proportion of biomass for the level of degradation, the authors ignore the effect of recovery time: a plot with a 10% loss of biomass would be considered the same as a plot that has lost 50% and recovered 40% of its biomass. The underlying hypothesis is that the processes determining species distribution are instantaneous. Although this may be a reasonable assumption

for many taxa, it may be problematic for some taxa (typically plants, because of their immobility and relatively long life spans). Have the authors tested the effect of recovery time in combination with biomass reduction on the descriptors of the occurrence curves (change points and maximum rate points)?

d) I was confused by Fig. S2B: if the random effect is on the intercept in GLMMs (which would translate into a different inflection point but a similar slope in the logistic function), why does the slope vary between curves?

e) I was curious of the reasons why some of the GLMMs that didn't converge. Was it because of a lack of data? Were there non-linear responses to biomass removal (e.g. higher occurrence levels at intermediate biomass decrease levels?)

f) Some of the methods are not fully described, e.g. the method for generating density curves, or how taxa were assigned to a functional group (functional trait thresholds, etc). In addition to describing these, it would be good to make the codes for producing all the results (from downloading the data from the zenodo repositories to producing the figures) available in a public repository to increase reproducibility. I would be happy to check the codes if necessary.

D. Appropriate use of statistics and treatment of uncertainties

(some of these points are mentioned in C)

E. Conclusions: robustness, validity, reliability & References: appropriate credit to previous work?

The study was carried out at a specific site with its own species and environmental constraints. The general conclusions on disturbance thresholds are therefore difficult to extrapolate to other tropical forests unless supported by other studies.

The results could be put into perspective with other studies in different tropical forests, even if they have worked with a smaller subset of taxa. Multi-site studies such as Burivalova et al. (2014; doi: 10.1016/j.cub.2014.06.065) could help to place the paper in a more general context. Otherwise, more caution should be exercised regarding the generalisability of the results.

Some results are at odds with previous scientific papers, and this should probably be discussed. For instance, the decrease in occurrence of most plant functional groups (Fig. 3A) contradicts the usual increase in low-density pioneer species after disturbance (e.g.: Carreño Rocabado et al. 2012, doi: 10.1111/j.1365-2745.2012.02015.x) [As you might have guessed, I work mostly with plant communities].

The term biodiversity (used throughout the text) should be used with more caution: the paper focuses on changes in the patterns of occurrence of different taxa with forest degradation (compared to old-growth forests), and biodiversity is not measured directly. Biodiversity patterns

can be quite decorrelated with composition changes: the composition could be completely changed but the biodiversity remains at a similar level.

Clarity and context: lucidity of abstract/summary, appropriateness of abstract, introduction and conclusions

I found the results difficult to understand from the main text alone, and a wider scientific community outside of ecology would have similar difficulties. This is mainly due to the introduction of many concepts that are specific to this study (change points, maximum rate points, vulnerability, etc) and are too briefly described in the main text and figure legends. I would therefore recommend the following changes to improve the clarity of the results:

a) A figure similar to Figure S2 A & C (which could be merged with Figure 1) in the main text illustrating the raw data (occurrences) and the GLM used; and an illustrated definition of change points and maximum rate points). I understand the interest in having this information in the supplementary methods, but in my opinion it is essential for a good understanding of the results.

b) In addition (or alternatively), the interpretation of these key concepts (change points and maximum rate points) and a brief justification for using them could be more fully developed in the main text. Similarly, the definition of the terms 'acceleration' and 'proactive' and 'reactive' thresholds in Figures 1C & D are unclear in the main text and their interpretation is therefore difficult without reading the methods. There is also a risk of confusion with change points and maximum rate points, which are all defined on the basis of first and second derivatives, so I would recommend a really didactic approach to ensure that all these concepts, from which the conclusions of the article are drawn, are clearly explained in the main text (with more complete methods for estimating them in the supplemental information).

c) Figure 3 seemed particularly complicated to me (and perhaps unnecessarily so). There is a lot of information on the same figure that is not organised in a way that guides the reader's eye. Some symbols could be changed to improve the clarity: the arrows in Figure 3A were a bit confusing as they don't give any directional information. The two types of triangles in Figure 3B are too similar, making it difficult to visualise patterns.

d) In Figure 3B, the use of the terms "probability of impact", "severity of impact", "sensitivity" and "vulnerability" should be harmonised. I would also mention that vulnerability is the product of probability and severity, in Figure 3 and elsewhere (instead of "combination", which is less accurate).

e) In Fig. 3, I had difficulty linking the values of change points in Fig. 3A and severity of impact in Fig. 3B (e.g. arboreal - all taxa: change point is close to zero, so I expected severity of impact to be close to 1, but severity of impact is close to 0.5). (See related comment C.b).

Other minor comments I had are listed below:

- Ref 1 is equivalent to Ref 31.
- Fig. S1: indicate in the legend that the bar length is on a log scale
- Fig. S2: The second derivative is the black line and the first derivative is the grey line.
- P 8: "which reinforces previous analyses showing that logged forests have higher ecosystem energy flows than primary forests": is there an underlying hypothesis that higher energy flows can support more species? Or the other way round, that a higher species richness leads to a higher niche occupancy and thus to higher energy flows? In any case, this should be made explicit.
- It could be added that, by definition of the logit link function, maximum rate points correspond to a 50% probability of occurrence (as an alternative interpretation).

Referee #2 (Remarks to the Author):

Thank you for the opportunity to review this manuscript. I very much enjoyed reading this, and I much liked the approach towards identifying change points and maximum rate points. These have immense relevance for conservation planning. The intensive effort that has gone into this study is fantastic, and the resulting taxonomic (and functional group) coverage impressive, and perhaps unparalleled anywhere in the world. The survey data combined with functional traits makes the information that goes into this study very comprehensive. The methods and choice of metrics are appropriate, statistical tests are suited to the questions asked and the figures informative and aesthetically appealing. The results are highly relevant to today's global tropical conservation landscape. While the value of logged forest for biodiversity has been often remarked upon in the past, this study is a huge advance over what we know already.

Some of my major concerns were:

1. While enormously impressive, the data ultimately come from a single site. A lot of recent work has shown that the same taxon can vary greatly in its response to habitat degradation based on, for instance, the abiotic factors prevalent at a site. (See, for example, Williams et al. 2021 *Global Change Biology*, 28, 797-815.) Therefore, while the taxonomic and functional generalisability at the SAFE project site might hold (although see comments below), the ecological thresholds (change points, maximum rate points, 30%, 68%) to inform conservation decisions might not be geographically generalisable, even for the same taxa. This needs some discussion. The authors' approach, however, is nonetheless valuable.

2. I am not sure that I quite agree with this statement:

"One-quarter of the taxa (n = 1,214) were detected in more than one survey, and more than half (54 %) of individual surveys consisted of multiple site visits (repeated observations of the same sites

within the survey year), limiting the potential for ecological context-dependence to influence our results". (Also, I believe that this is three-quarters of the taxa analysed?)

If each of the surveys were conducted in a single year, year-specific context dependence (from climate, natural fluctuations in population size, etc.) is still a problem, especially for the 25% of taxa that were detected in only a single survey. This would be especially true if populations cycles of the same species are not synchronised between primary and degraded forest, and therefore follow different trajectories across various habitat types. Would it be possible to show that for the species that were detected across multiple surveys, differences in occurrence patterns between primary and degraded forest were consistent across years?

This would then provide some evidence for the statement that "...based on the statistical assumption that a large sample size of taxon responses widely distributed across the tree of life will generate a sample average that closely reflects the community-level mean response". While I don't immediately see how this study overcomes potential problems with single-survey results, the data are nonetheless impressive enough, most of the results consistent with prior work and the identification of change points and maximum rate points an important advance. I would suggest jettisoning the "individual survey" aspect of this study.

3. I have the same comment (as above) regarding this:

"Data sources that sampled multiple years were split into separate, annual surveys, allowing us to more accurately align biodiversity observations with forest degradation measurements taken at different time points, and to account for year-to-year variation in taxon specific responses to the same ecological gradient."

It would be important to see the distribution of the number of surveys on which the 1,214 taxa was recorded. If the majority of these taxa were recorded in just two or three surveys, for instance, the problem of year-specific context-dependence might still be an issue, I think.

Some minor comments:

1. The abstract reads a little generic and does not explicitly reflect the novelty of this study, which is the identification of thresholds to guide conservation decisions, as stated in the introduction. As of now, the abstract largely feels like a repetition of what we already know from the literature (e.g., logging harms large species, specialists, etc.). I would suggest highlighting the truly novel aspect of this work in as many words.

2. Single year comparisons between primary and degraded forest also suffer from the fact that the confounding impacts of climate change are ignored. See, for instance, the paper on shifted baselines from the Amazon (Stouffer et al. 2021 *Ecology Letters*, 24, 186-195) which shows that even in primary forest, climate impacts (most likely) changes community composition over time, with certain functional groups more vulnerable than others. If the pace at which climate change affects biodiversity in primary and logged forest is different, then snapshot comparisons of biodiversity in the two habitat types might not reflect future equilibrium states of community composition. While this is difficult or perhaps even impossible to address without very long-term data, I believe it certainly merits acknowledgment and discussion.

3. The main text states that one-quarter of taxa were recorded in more than one survey but the methods state that three-quarters of taxa were recorded in more than one survey. The latter is true? There are some discrepancies in proportions and numbers of taxonomic groups analysed, etc. Please do make a thorough check for these.

Referee #3 (Remarks to the Author):

This study aims at estimating thresholds of habitat loss for bioconservation in tropical forest. As the authors admit such thresholds have been assessed for numerous single taxa. This study uses a multitaxon approach based on impressive data from Sabah, Malaysia. The major result is to provide a synthetic view on biodiversity effects for a single habitat type, tropical forest. For experts in the field, the results do not come to a surprise given the extensive prior work. From a technical perspective the submission looks sound, the methods are appropriate and sufficiently described. All raw data are available.

According to this study a change in land-use (logging of primary forest) by about 30% seems safe with respect to biodiversity and functional loss, while above 68% loss severe consequences set in. This does not mean that single taxa might react different. Because ecological functioning is impacted by the position in trophic networks it would have been interesting to get more information about the logging impact for different trophic levels. The diet and trophic levels categories in Figs 2 and 3 are not sufficient in this respect. I guess that logging particularly impacts taxa at higher trophic levels (larger predators).

Because the paper can be read as advocating that some degree of logging (30%?) should be save, it might have broader political implication. I'm not sure whether the authors are fully aware of this because political or economic implications are not mentioned. For a sound discussion the results must stand scrutiny at the taxon level. They are potentially in conflict with earlier work on biodiversity loss due to habitat loss and have to differentiate between logging types. Salvage logging is qualitatively different from logging for palm oil plantations.

I missed more detailed information about the fate of the logged areas. The methods section only tells about a mix of fates, partly ending into palm oil plantations. The latter are known to retain a considerable biodiversity of birds if undisturbed forest is near. Salvage logging areas should also reduce biodiversity effects. In turn, the transformation of primary forest into open landscapes should have the most severe impact on diversity. Recalculating these different practices into a single metric, might miss this differences. From the methods section it did not became clear whether a differentiation into logging practices would be possible. However, for a paper that intends to provide guidelines I expected to see such reference to logging practice. In this respect a direct multitaxon comparison of primary forest and palm oil plantation might have been more informative.

I also missed the temporal perspective. Biodiversity effects after habitat destruction often come with a time lag. Therefore, single annual surveys might not catch the silent biodiversity loss in time.

Much work on biodiversity loss after habitat reduction/transformation has focused on the species area relationship (SAR). The present degradation metric is related to the area variable of the SAR (biomass loss instead of area loss, both should be positively correlated). Given a power function SAR with slope of 0.5 (commonly found in small scale surveys) would retain 84% of species at 30% habitat loss. A slope of 0.3 predicts 90% species remaining. At 70% habitat loss and a slope of 0.5

only 55% of species would remain. That's comparable to the present result. I think the relationship to the SAR approach needs to be discussed, maybe with pros and cons of both methods. How strong does biomass correlate with area?

Ewers et al. warn that single year snapshots might give highly inaccurate biodiversity patterns. This is surely true. The present paper tries to tackle this problem by the sheer mass of data and the assumption of statistical averaging. Nevertheless a quarter of data still relies on such single snapshots. All surveys were done in a single year. If these data were autocorrelated, for instance due to similar habitat and weather conditions during the respective study year, the results still would be biased. I think the authors need to clarify this point and present more information on when the surveys were done, whether multiple taxa were observed during such single surveys, and how large were the degrees of collinearity and of spatial autocorrelation.

At the end of page 6 it is written "Of these, 1,681 taxa and all 126 functional groups were able to be modelled individually (≥ 5 occurrences)." I'm not sure what 5 occurrences means. Does this refer to 5 different sampling sites across the logging gradient? If yes how did you assess the sensitivity to logging for such small sample sizes? In other words, would a different abundance cut-off give different results?

Beginning of line 7. Why is the area 'experimental'? Are there artificially altered forest parts for scientific studies?

The data represent a wide number of taxa and functional types. However, they are still heavily biased with respect to biodiversity. Insects, spiders, nematodes, or molluscs are highly underrepresented, while the focus is on vertebrates and part of plants. This bias should have been discussed.

The caption of Fig. 3B is insufficient. The symbols are labelled as probability and severity, while the caption names susceptibility and sensitivity. Anyway the symbols needs clarification as captions should be self-explaining.

Author Rebuttals to Initial Comments:

Referee #1 (Remarks to the Author):

A. Summary of the key results

This study presents an interesting approach to the multi-dimensional effects of tropical forest degradation (and in particular selective logging) on forest composition, at both taxonomic and functional levels. To do so, it analyses how species occurrence patterns change within a gradient of forest degradation. The authors provide thresholds of biomass removal above below which degraded forests could significantly contribute to biodiversity conservation, and above which logged forests would need restoration actions to maintain high diversity levels. The conclusions should be of interest not only to ecologists, but also to policy makers and forest managers seeking to establish general rules for biodiversity conservation in anthropised forest landscapes.

Thank you! We're glad you think this manuscript will be of interest and impact to academics and policy makers, and that the key results we've presented have come through clearly.

B. Originality and significance: if not novel, please include reference

The originality of this study lies in the use of a large number of taxa across a wide range of life forms, from plants to insects, mammals, etc., which to my knowledge has never been done before.

Thank you. It has been a massive effort from hundreds of people to generate this diversity of data, and we're pleased you have recognised the unique nature of the data we present.

C. Data & methodology: validity of approach, quality of data, quality of presentation

a) As defined, the probability of impact could depend on the sampling effort (e.g. the number of observations) in different taxonomic and functional groups. Is there a link between these two variables?

The unit of replication (sample size) for the probability of impact metric is the number of taxa per taxonomic or functional group. You ask an important question, so we have now checked and there is no significant correlation between number of taxa and probability of impact for either taxonomic groups ($r = -0.21$, $df = 8$, $p = 0.56$) or for functional groups ($r = -0.11$, $df = 45$, $p = 0.45$).

We have added the following statement about this in the Methods (L760): “Correlation analyses demonstrated that there was no impact of sample size (the number of taxa per group) on probability of impact for either taxonomic groups ($r = -0.21$, $df = 8$, $p = 0.56$) or for functional groups ($r = -0.11$, $df = 45$, $p = 0.45$).”

b) I suggest using the median rather than the mean to estimate impact severity, unless there is a good reason to use the mean. The median is less sensitive to outliers, and the effect of outliers could explain some discrepancies between impact severity and taxa change points in Fig. 3.

You raise a good point and we would normally agree with you. The problem we faced is that Impact Severity values are both bounded and have a highly variable distribution. To give an example: there were six Arachnid taxa with turning points of 100, but 22 Arachnid taxa with turning points of 0. The median for Arachnids was therefore zero, but the mean was 29. Faced with data distributions of this nature, we felt the mean gives a better indication of the spread of values than the median.

c) By substituting the proportion of biomass for the level of degradation, the authors ignore the effect of recovery time: a plot with a 10% loss of biomass would be considered the same as a plot that has lost 50% and recovered 40% of its biomass. The underlying hypothesis is that the processes determining species distribution are instantaneous. Although this may be a reasonable assumption for many taxa, it may be problematic for some taxa (typically plants, because of their immobility and relatively long life spans). Have the authors tested the effect of recovery time in combination with biomass reduction on the descriptors of the occurrence curves (change points and maximum rate points)?

Thank you for raising this, and you're right to suggest we should give this more attention. We agree that the metric combines these two processes of biomass loss and gain, to give a summary metric that represents the present state of the forest. We believe this is the most relevant metric to focus our analyses on, as present-day biomass is typically the easiest, and most reliable, metric for managers to gain access to. The majority of individual sampling sites in our study area, for example, have been through anywhere from one to four cycles of logging and recovery (Struebig et al. 2013, Riutta et al. 2018). None of the recovery periods were long enough for the forest to fully recover the biomass that was removed in any of the logging rounds, meaning there is no obviously defensible metric of “recovery time” for sites with a complex logging history.

We now draw readers attention to this issue in the Methods, where we state (L580): “This metric implicitly combines the initial removal of woody biomass through logging and land clearance with the gradual recovery of biomass that may have occurred since the last disturbance event, meaning our metric of forest degradation reflects the present-day balance between these two opposing forces.” Furthermore, in the main text we highlight the possibility that the processes and patterns governing forest recovery may be different to those arising from forest loss (L328): “We note, however, that our analysis examines the transition from unlogged to logged forest, and that effectively reversing that transition may require restoring the biomass of a forest beyond this transition point.”

We also draw further attention to the issue in a new paragraph in the Methods, that also addresses a concern about shifting baselines raised by Reviewer 2 (L709):

*“Long-term shifts in the composition of forest communities might mean the biodiversity patterns we associate with primary forest in our data are themselves depauperate relative to historical patterns (Stouffer et al. 2021). Similarly, the complex logging history of our study site with repeated, but unequally distributed, rounds of logging means many sites have been through multiple stages of degradation separated by partial recovery (Struebig et al. 2013, Riutta et al. 2018). Our data are not sufficient to quantify historical patterns of occupancy nor the impact of time lags on trajectories of occupancy, so we are unable to directly test for these effects. Nonetheless, long term declines and local extinction of megafauna like the Sumatran rhino *Dicerorhinus sumatrensis harrissoni* (Kretzschmar et al. 2016) make it likely that a shifting baseline is a valid concern at our study site. However, we have no way of knowing whether the rates of biodiversity change from the processes that might generate baseline shifts will be the same or different in primary and logged forest. Consequently, we can only emphasise that our analyses are based on a space-for-time substitution, which makes the implicit assumption that the effects of habitat degradation we quantify are additional to, and do not interact with, any other processes contributing to long-term biodiversity change.”*

d) I was confused by Fig. S2B: if the random effect is on the intercept in GLMMs (which would translate into a different inflection point but a similar slope in the logistic function), why does the slope vary between curves?

Apologies for the confusion. The different curves represent survey-specific linear models (models fitted to that year alone, for which there are no random effects). It is a better illustration of variance in the raw data than presenting just the final fitted, random effects

model where, as you state, all surveys would have the same slope. We have clarified this in the caption (L657): “Some taxa were observed in multiple surveys (represented by semi-transparent lines, here fitted as survey-specific linear models), each of which could have a different occurrence pattern (Ewers et al. 2024). In these cases, we used a mixed effect model to combine observations across all datasets, generating a single model of that taxon’s occurrence pattern that was used to determine turning and maximum rate points (thick line).”

e) I was curious of the reasons why some of the GLMMs that didn’t converge. Was it because of a lack of data?

This is a good point that we should have clarified in the text. Most GLMMs did converge. In the Methods, we had stated “We were able to fit GLMMs to 798 taxa (47 % of fitted models)”, but as a take-home value this understates how effective the fitting process was. Only 56 % of taxa were observed in multiple studies and so we were only able to attempt fitting GLMMs to those 946 taxa, meaning successful fits were obtained in 84 % of cases. Where GLMMs did fail to converge, it was almost certainly driven by data limitations. Specifically, taxa observed in multiple datasets weren’t necessarily observed equally in all datasets, and low numbers of observations in one or more surveys can limit the ability of a GLMM to estimate survey-specific random effects.

We have added the following text to the Methods to clarify this (L639): “We were able to fit GLMMs to 798 out of the 946 taxa that were observed in multiple studies (84 % of fitted models) and 72 functional groups (59 %). The main reason by which GLMMs failed to converge was because taxa or functional groups observed in multiple datasets weren’t necessarily observed equally in all datasets, and low numbers of observations in one or more surveys can limit the ability of a GLMM to estimate survey-specific random effects.”

Were there non-linear responses to biomass removal (e.g. higher occurrence levels at intermediate biomass decrease levels?)

Thank you for raising this; it’s something we had examined and then discarded in exploratory analyses, but hadn’t described in the manuscript. We had initially explored the possibility of intermediate peaks/troughs by incorporating a second-order polynomial term in the models, and in a small number of instances models with significant polynomial terms were selected as the best model. However, in all cases where we went to visually inspect the raw data and

model diagnostic plots, we discovered the inclusion of the polynomial term was questionable. Commonly, a 'peak' would be fitted because of a cluster of presence points at either a very high or very low value of biomass removal were accompanied by a single absence point, resulting in a very slight reduction of modelled occurrence probability at one extreme or the other. In no instances did we observe a polynomial model that smoothly in/decreased and then de/increased along the degradation gradient. We therefore omitted polynomial terms from our model fitting process.

We have added a statement about this to the Methods (L633): "We tested for a main, linear effect of forest degradation only. This was because visual inspection and diagnostic plots of exploratory analyses containing a polynomial term failed to identify clear cases of taxa that had peaks in occurrence at intermediate levels of biomass removal."

f) Some of the methods are not fully described, e.g. the method for generating density curves, or how taxa were assigned to a functional group (functional trait thresholds, etc).

Apologies for these gaps. We have now added additional information about the calculation of density curves to the "Identifying thresholds" section of the Methods, naming the specific function and settings used to calculate them (L725): "Density curves were fitted using the kernel density estimation function with default settings in the 'stats' package (R Development Core Team 2021)."

We have also revised the relevant section in the Methods to make clearer how functional traits were categorized. For traits categorised into two groups, we divided taxa according to whether they were above or below the median value for that trait (L548): "which we categorised into two groupings for analysis (low and high according to whether trait values were below or above the median respectively)." Body mass was the only trait divided into three groups. We now describe this process in the Methods as (L553): "Body mass was categorised into three groupings (low, medium and high) separately for each taxonomic group. Grouping boundaries were set by \log_{10} -transforming body mass and dividing taxa into three equal quantiles."

In addition to describing these, it would be good to make the codes for producing all the results (from downloading the data from the zenodo repositories to producing the figures) available in a public repository to increase reproducibility. I would be happy to check the codes if necessary.

We have made the code available at <https://github.com/robewers01/SAFE-thresholds>. Should the MS be accepted, we will generate a new, clean repository that we will cite in the MS itself.

D. Appropriate use of statistics and treatment of uncertainties

(some of these points are mentioned in C)

We hope you agree that we've addressed all of the relevant points about statistics in our responses above. To summarise, we provided additional analyses and text about the potential impact of sample size on the Probability of Impact metric; the use of means versus medians for calculating Impact Severity; the potential impact of recovery time and how it relates to our metric of forest degradation; clarification about our use of GLMMs, their convergence rate and reasons for why some failed to converge; the reason for examining linear effects only; additional details about the calculation of density curves and the categorisation of functional traits; and we have provided you with access to all of the analysis code.

E. Conclusions: robustness, validity, reliability & References: appropriate credit to previous work?

The study was carried out at a specific site with its own species and environmental constraints. The general conclusions on disturbance thresholds are therefore difficult to extrapolate to other tropical forests unless supported by other studies. The results could be put into perspective with other studies in different tropical forests, even if they have worked with a smaller subset of taxa. Multi-site studies such as Burivalova et al. (2014; doi: 10.1016/j.cub.2014.06.065) could help to place the paper in a more general context. Otherwise, more caution should be exercised regarding the generalisability of the results.

We agree with you entirely that putting our results into context is important. We have chosen to place our results in the context of the High Carbon Stock Approach threshold for defining forest thresholds, given this is a globally applied standard used in tropical forest management (L309): "This value is similar to the more arbitrary definition of a high density forest in the widely used High Carbon Stock Approach (Rosoman et al. 2017), which sets a threshold at 150 t.ha⁻¹ of carbon regardless of pre-logging biomass (equivalent to 25 % biomass reduction at our study site)."

We have also added a new statement related to the generalisability of our results to the last section of the MS (L425): "Our data were collected from a single site, however, and taxon responses to habitat degradation can vary across geographical gradients (Orme et al. 2019,

Williams and Newbold 2021), so more studies of a similar nature will be required to strengthen confidence in the generality of our conclusions.”

We cite the important paper by Burivalova et al. (2014) twice in our MS (L195, 456), but do not believe it is valid to make a more specific comparison. This is for two reasons. First, they present logging impacts in units of $m^3 ha^{-1}$ of timber extracted, but this is a very different metric to what we analyse, which is the total biomass lost through the logging process. There is no direct correlation between these two metrics, because trees vary in wood density within and among forests, and because collateral damage can destroy biomass above and beyond that which is extracted. This damage varies a huge amount among logging systems, and can account for more than double the amount of extracted biomass (Pinard and Putz 1996). It is therefore not possible to confidently compare Burivalova’s values with ours. Second, Burivalova highlighted a threshold point where 50% of species might be lost, but they only presented this for taxa that lost more than 50% of their species (i.e. mammals and amphibians). They did not present an equivalent value for taxa that had losses lower than this (invertebrates) or that became more species rich (birds). Consequently, the quantitative, threshold values they provide represent a worst-case scenario, while we have used the breadth of sampling across all taxa and all responses to understand impacts at a community level.

Some results are at odds with previous scientific papers, and this should probably be discussed. For instance, the decrease in occurrence of most plant functional groups (Fig. 3A) contradicts the usual increase in low-density pioneer species after disturbance (e.g.: Carreño Rocabado et al. 2012, doi: 10.1111/j.1365-2745.2012.02015.x) [As you might have guessed, I work mostly with plant communities].

Thank you for raising this – it’s an important point for us to explain. This apparent contradiction arises because of the magnitude of the logging intensity gradient we examine in our study. The only way to extract such large proportions of biomass from a diverse tropical forest is to extract individuals that are much smaller, and belong to a much wider range of timber species, than is normal for selective tropical logging. In practice, this means extracting individuals and species of lower and lower commercial value, which broadly correlates with wood density and the boundary between ‘climax’ and ‘pioneer’ species. Taken to the extreme – habitat conversion where all trees are removed – the occurrence probability of all species, including pioneers, must drop to zero.

Most other studies of selective logging are from sites with far more restricted extraction rates. For example, the most heavily logged forest treatment in the Carreño Rocabado (2012) study that you reference had lost just 4 trees ha⁻¹ in a location where average stem density was 368 ha⁻¹. By contrast, some of the most extreme plots at our site had as much as 80 % of their stems removed (Pfeifer et al. 2015). With such dramatic reductions in the number of individual trees, it should be expected that even pioneer species should have reduced occurrence probabilities when examined along the full breadth of the forest disturbance gradient.

We have added an explanation of this effect to the MS, citing the specific paper the reviewer has mentioned (L397): “Pioneer tree species, including those with low wood density, might normally be expected to increase rather than decrease in response to logging disturbance (Carreño-Rocabado et al. 2012). However, progressively increasing amounts of biomass removal necessarily results in the extraction of a progressively higher proportion of standing trees (Pfeifer et al. 2015), which necessarily includes species with low wood density.”

The term biodiversity (used throughout the text) should be used with more caution: the paper focuses on changes in the patterns of occurrence of different taxa with forest degradation (compared to old-growth forests), and biodiversity is not measured directly. Biodiversity patterns can be quite decorrelated with composition changes: the composition could be completely changed but the biodiversity remains at a similar level.

Biodiversity is a term with varied usage and definitions across the scientific literature. While we don't believe we have used the term in a way that is inconsistent with common usage, we have nonetheless gone through the text and removed it from a number of places. As examples, we have updated several statements in the Abstract in the following manner (L168): “using one of the most comprehensive assessments of ~~biodiversity~~ taxon responses to habitat degradation in any tropical forest environment,” and (L172) “with the impacts on ~~biodiversity~~ individual taxa and functional groups seen at very low levels of logging impact.” Throughout, we have avoided referring to our own analyses as examining biodiversity, but we have retained the term when referring to the wider literature.

Clarity and context: lucidity of abstract/summary, appropriateness of abstract, introduction and conclusions

I found the results difficult to understand from the main text alone, and a wider scientific community outside of ecology would have similar difficulties. This is mainly due to the introduction of many

concepts that are specific to this study (change points, maximum rate points, vulnerability, etc) and are too briefly described in the main text and figure legends. I would therefore recommend the following changes to improve the clarity of the results:

a) A figure similar to Figure S2 A & C (which could be merged with Figure 1) in the main text illustrating the raw data (occurrences) and the GLM used; and an illustrated definition of change points and maximum rate points). I understand the interest in having this information in the supplementary methods, but in my opinion it is essential for a good understanding of the results.

This is a good idea thank you. We have added small insets into the panels of Fig 1C and D to illustrate how change and maximum rate points are calculated.

b) In addition (or alternatively), the interpretation of these key concepts (change points and maximum rate points) and a brief justification for using them could be more fully developed in the main text. Similarly, the definition of the terms 'acceleration' and 'proactive' and 'reactive' thresholds in Figures 1C & D are unclear in the main text and their interpretation is therefore difficult without reading the methods. There is also a risk of confusion with change points and maximum rate points, which are all defined on the basis of first and second derivatives, so I would recommend a really didactic approach to ensure that all these concepts, from which the conclusions of the article are drawn, are clearly explained in the main text (with more complete methods for estimating them in the supplemental information).

These are all good points, and we have worked hard to ensure they are clarified in the text. We have made the following three changes:

- 1. We have added more background to the concepts of proactive and reactive conservation in the Introduction (L199): "Conservation actions globally can be largely categorised as being either proactive or reactive (Brooks et al. 2006). Proactive conservation targets areas of low vulnerability, and relatively passive approaches such as protecting the habitat are expected to deliver positive outcomes for biodiversity. By contrast, reactive conservation targets areas of high threat, where immediate action is required to stave off biodiversity loss." We have also removed these terms from the caption to Fig. 1, as they introduced unnecessary complexity in that context.*
- 2. We noticed we had introduced unnecessary confusion by using the term 'accelerate' in two contexts: while describing patterns of occurrence for individual*

taxa (e.g. Fig. S3); and while describing the accumulation rate of impacted taxa (e.g. Fig. 1). To clarify this, we now only use acceleration in the latter context, and have reworded the former to avoid this term.

- 3. We have added a new paragraph to the main text to introduce the concept of change and maximum rate points, and how they are calculated (L247): “We focus our analyses on two critical points in the responses of individual taxa to habitat degradation. We term the first a “change point”, which we define as the first point along the degradation gradient at which a taxon exhibits a discernible change in occurrence probability. The second follows the first and we term it a “maximum rate point”. These represent the point along the forest degradation gradient where the rate of change in occurrence probability is the most rapid. Both change and maximum rate points were calculated from derivatives of the fitted occurrence models (Methods; Fig. S3).”*

c) Figure 3 seemed particularly complicated to me (and perhaps unnecessarily so). There is a lot of information on the same figure that is not organised in a way that guides the reader's eye. Some symbols could be changed to improve the clarity: the arrows in Figure 3A were a bit confusing as they don't give any directional information. The two types of triangles in Figure 3B are too similar, making it difficult to visualise patterns.

Thank you for these helpful comments. We have now replaced the arrowheads in Fig 3B with a small, filled circle, and one of the triangles in Fig. 3B with a circle.

d) In Figure 3B, the use of the terms "probability of impact", "severity of impact", "sensitivity" and "vulnerability" should be harmonised. I would also mention that vulnerability is the product of probability and severity, in Figure 3 and elsewhere (instead of "combination", which is less accurate).

Thank you for spotting these inconsistencies in our terminology. We've the legend and caption of Fig 3 accordingly (L390), and also discovered a similar problem in the Methods which we have also corrected (L769, 786).

e) In Fig. 3, I had difficulty linking the values of change points in Fig. 3A and severity of impact in Fig. 3B (e.g. arboreal - all taxa: change point is close to zero, so I expected severity of impact to be close to 1, but severity of impact is close to 0.5). (See related comment C.b).

This is because different sets of taxa are used in the calculations for Figs 3A and 3B. For the particular group you mention – all arboreal taxa – there are 127 taxa combined and modelled with a GLMM to generate the single change point for that functional group. This is what gets presented in Fig. 3A. By contrast, when those 127 taxa were modelled individually, we were only able to estimate turning points for 61 of them, and so the values in Fig. 3B are calculated from this subset of the taxa.

Other minor comments I had are listed below:

- Ref 1 is equivalent to Ref 31.

Apologies for this error and thank you for picking it up. We've now updated the citation list accordingly.

- Fig. S1: indicate in the legend that the bar length is on a log scale

We've added this information as requested.

- Fig. S2: The second derivative is the black line and the first derivative is the grey line.

Thank you! We've amended the caption accordingly.

- P 8: "which reinforces previous analyses showing that logged forests have higher ecosystem energy flows than primary forests": is there an underlying hypothesis that higher energy flows can support more species? Or the other way round, that a higher species richness leads to a higher niche occupancy and thus to higher energy flows? In any case, this should be made explicit.

We're not aware of any explicit hypothesis of this nature, although it is consistent with the data presented by Malhi et al. (2022). We have added Malhi's result about increased species richness to this sentence (L264): "which reinforces previous analyses showing how logged forests have higher ecosystem energy flows and higher species richness than primary forest."

- It could be added that, by definition of the logit link function, maximum rate points correspond to a 50% probability of occurrence (as an alternative interpretation).

Good idea thank you. We have added this to the Methods (L674): “This point was numerically estimated by identifying the point at which the predicted occurrence pattern from the binomial GLM had the highest absolute slope (as represented by the root of the second derivative), and corresponds to the point along the habitat degradation gradient where the probability of occurrence is 50 %.”

Referee #2 (Remarks to the Author):

Thank you for the opportunity to review this manuscript. I very much enjoyed reading this, and I much liked the approach towards identifying change points and maximum rate points. These have immense relevance for conservation planning. The intensive effort that has gone into this study is fantastic, and the resulting taxonomic (and functional group) coverage impressive, and perhaps unparalleled anywhere in the world. The survey data combined with functional traits makes the information that goes into this study very comprehensive. The methods and choice of metrics are appropriate, statistical tests are suited to the questions asked and the figures informative and aesthetically appealing. The results are highly relevant to today’s global tropical conservation landscape. While the value of logged forest for biodiversity has been often remarked upon in the past, this study is a huge advance over what we know already.

Thank you. We were really encouraged to read such positive comments.

Some of my major concerns were:

1. While enormously impressive, the data ultimately come from a single site. A lot of recent work has shown that the same taxon can vary greatly in its response to habitat degradation based on, for instance, the abiotic factors prevalent at a site. (See, for example, Williams et al. 2021 *Global Change Biology*, 28, 797-815.) Therefore, while the taxonomic and functional generalisability at the SAFE project site might hold (although see comments below), the ecological thresholds (change points, maximum rate points, 30%, 68%) to inform conservation decisions might not be geographically generalisable, even for the same taxa. This needs some discussion. The authors’ approach, however, is nonetheless valuable.

Thank you – you’re right to raise this as an important caveat to our results. We have now added a statement outlining this concern to the last section of the MS (L425): “Our data were collected from a single site, however, and taxon responses to habitat degradation can vary

across geographical gradients (Orme et al. 2019), so more studies of a similar nature will be required to strengthen confidence in the generality of our conclusions.”

2. I am not sure that I quite agree with this statement:

“One-quarter of the taxa (n = 1,214) were detected in more than one survey, and more than half (54 %) of individual surveys consisted of multiple site visits (repeated observations of the same sites within the survey year), limiting the potential for ecological context-dependence to influence our results”. (Also, I believe that this is three-quarters of the taxa analysed?)

Apologies for this confusion. It arose because we were interchanging numbers from two totals: one was the total of all taxa we had data on, and the other from the subset total of taxa that we were able to model. We have now updated the values throughout the main MS and Methods section to focus on only the taxa that were modelled. There were 1,681 taxa successfully modelled, and of this subset, 946 (56%) were observed in multiple datasets.

If each of the surveys were conducted in a single year, year-specific context dependence (from climate, natural fluctuations in population size, etc.) is still a problem, especially for the 25% of taxa that were detected in only a single survey. This would be especially true if populations cycles of the same species are not synchronised between primary and degraded forest, and therefore follow different trajectories across various habitat types. Would it be possible to show that for the species that were detected across multiple surveys, differences in occurrence patterns between primary and degraded forest were consistent across years?

You raise a good question, and it is the subject of a related analysis and manuscript that we made available to reviewers as a Supporting MS. We have now uploaded this MS to BioRxiv to ensure it is accessible to all readers (Ewers et al. 2024). In short, the answer is no: occurrence patterns of any given taxon are highly variable among years. That is why we have gone to such lengths to give a clear description of the density of repeat sampling in this current MS.

This would then provide some evidence for the statement that “...based on the statistical assumption that a large sample size of taxon responses widely distributed across the tree of life will generate a sample average that closely reflects the community-level mean response”. While I don’t immediately see how this study overcomes potential problems with single-survey results, the data are nonetheless impressive enough, most of the results consistent with prior work and the identification of change

points and maximum rate points an important advance. I would suggest jettisoning the “individual survey” aspect of this study.

The vast majority of studies in this field would not even mention the issue of snapshot data, but based on our detailed analyses of among-survey variation in taxon responses (Ewers et al. 2024), we believe it would be misleading to not mention the general issue. To address your concern, we have now removed our initial claims that our data are complete enough to overcome the problem of context-dependence in biodiversity responses, including the sentence quoted above.

We need to retain the use of individual surveys for analysis, as that provides the most accurate mapping between biodiversity surveys and the state of the forest. Large parts of our landscape were logged during the data collection period we are analysing, for example, meaning they have different values of degradation at different points in the 11-year period during which we collected data.

3. I have the same comment (as above) regarding this:

“Data sources that sampled multiple years were split into separate, annual surveys, allowing us to more accurately align biodiversity observations with forest degradation measurements taken at different time points, and to account for year-to-year variation in taxon specific responses to the same ecological gradient.” It would be important to see the distribution of the number of surveys on which the 1,214 taxa was recorded. If the majority of these taxa were recorded in just two or three surveys, for instance, the problem of year-specific context-dependence might still be an issue, I think.

We have generated this figure as you requested (see below), and have added it to the Methods as Fig. S2. Of the 1,681 taxa that were modelled, 731 (44 %) were represented in a single survey, and the remaining 946 (56 %) were represented in multiple surveys.

In response to the your previous comment, we had removed the phrase about year-to-year variation, along with all claims that our analyses are not prone to context-dependence, which also helps address this present comment.

Some minor comments:

1. The abstract reads a little generic and does not explicitly reflect the novelty of this study, which is the identification of thresholds to guide conservation decisions, as stated in the introduction. As of now, the abstract largely feels like a repetition of what we already know from the literature (e.g., logging harms large species, specialists, etc.). I would suggest highlighting the truly novel aspect of this work in as many words.

This is a good point – thank you for raising it. We have now reduced the space dedicated to describing the functional changes in logged forest, and expanded our presentation of the thresholds which is the more novel aspect of our study.

2. Single year comparisons between primary and degraded forest also suffer from the fact that the confounding impacts of climate change are ignored. See, for instance, the paper on shifted baselines from the Amazon (Stouffer et al. 2021 Ecology Letters, 24, 186-195) which shows that even in primary forest, climate impacts (most likely) changes community composition over time, with certain functional groups more vulnerable than others. If the pace at which climate change affects biodiversity in primary and logged forest is different, then snapshot comparisons of biodiversity in the two habitat types might not reflect future equilibrium states of community composition. While this is difficult or perhaps even impossible to address without very long-term data, I believe it certainly merits acknowledgment and discussion.

As you acknowledge, addressing this through data verges on the impossible and we do not believe our data are sufficient to tackle this challenge. We have, though, added a new section to the Methods with the subheading “Temporal bias in results” in which we present the potential for shifting baselines to influence our data (L709):

*“Long-term shifts in the composition of forest communities might mean the biodiversity patterns we associate with primary forest in our data are themselves depauperate relative to historical patterns (Stouffer et al. 2021). Similarly, the complex logging history of our study site with repeated, but unequally distributed, rounds of logging means many sites have been through multiple stages of degradation separated by partial recovery (Struebig et al. 2013, Riutta et al. 2018). Our data are not sufficient to quantify historical patterns of occupancy nor the impact of time lags on trajectories of occupancy, so we are unable to directly test for these effects. Nonetheless, long term declines and local extinction of megafauna like the Sumatran rhino *Dicerorhinus sumatrensis harrissoni* (Kretzschmar et al. 2016) make it likely that a shifting baseline is a valid concern at our study site. However, we have no way of knowing whether the rates of biodiversity change from the processes that might generate baseline shifts will be the same or different in primary and logged forest. Consequently, we can only emphasise that our analyses are based on a space-for-time substitution, which makes the implicit assumption that the effects of habitat degradation we quantify are additional to, and do not interact with, any other processes contributing to long-term biodiversity change.”*

3. The main text states that one-quarter of taxa were recorded in more than one survey but the methods state that three-quarters of taxa were recorded in more than one survey. The latter is true? There are some discrepancies in proportions and numbers of taxonomic groups analysed, etc. Please do make a thorough check for these.

Apologies for this: we have clarified the numbers of taxa involved in our response to your comment 2 above. There were 1,681 taxa successfully modelled, and of this subset, 946 (56%) were observed in multiple datasets.

Referee #3 (Remarks to the Author):

This study aims at estimating thresholds of habitat loss for bioconservation in tropical forest. As the authors admit such thresholds have been assessed for numerous single taxa. This study uses a multitaxon approach based on impressive data from Sabah, Malaysia. The major result is to provide a synthetic view on biodiversity effects for a single habitat type, tropical forest. For experts in the field, the results do not come to a surprise given the extensive prior work. From a technical perspective the submission looks sound, the methods are appropriate and sufficiently described. All raw data are available.

Thank you for reviewing this manuscript, and for these positive comments.

According to this study a change in land-use (logging of primary forest) by about 30% seems safe with respect to biodiversity and functional loss, while above 68% loss severe consequences set in. This does not mean that single taxa might react different. Because ecological functioning is impacted by the position in trophic networks it would have been interesting to get more information about the logging impact for different trophic levels. The diet and trophic levels categories in Figs 2 and 3 are not sufficient in this respect. I guess that logging particularly impacts taxa at higher trophic levels (larger predators).

Thank you for drawing attention to this; our explanation should have been clearer. We divided taxa into 126 functional groups that encompassed a total of six trophic levels and 21 diet categories, which are all described in Table S2, and we used Figs 2 and 3 to summarise the key results from analysing all of these various combinations of functional categories. Fig. 3 only presents results for functional groups where we detected a significant impact of forest degradation: all “missing” groups had non-significant results. We have now clarified this in the Fig. 3 caption (L383): “Analyses were conducted on the 126 functional groups described in Table S2, but here we present only functional groups that had statistically significant responses to forest degradation. All other groups not displayed had non-significant responses to degradation..”

With respect to your specific question about predators, we found no significant effect in this functional group and this is why they are not represented in Fig 3. We have added a specific statement to this effect to the MS (L376): “We found no general pattern with respect to trophic level, with no evidence that predators were more susceptible to habitat degradation than herbivores.”

Because the paper can be read as advocating that some degree of logging (30%?) should be saved, it might have broader political implications. I'm not sure whether the authors are fully aware of this because political or economic implications are not mentioned.

You're right that we should have made these implications more explicit. The goal of this MS was to identify policy-relevant thresholds in logging damage that decision makers might care about, but the political and economic implications of applying any of these thresholds will always be specific to the particular location being managed, making it difficult to generalise a statement about these issues.

We have settled on making specific reference to issues of social equity when making decisions linked to the thresholds we present (L430): "Proactive conservation decisions – actions designed to safeguard a habitat against further degradation – in these relatively lightly degraded forests could include adding them directly to the conservation estate by giving them protected area status (Reynolds et al. 2011), should that be a valid and equitable approach to conservation in the region (Schultz et al. 2022)." We have also added a phrase to highlight the importance of local political and economic implications (L433): "depending on the local political and economic situation, maximum timber extraction rates could be set at levels that ensure the threshold is not passed."

For a sound discussion the results must stand scrutiny at the taxon level. They are potentially in conflict with earlier work on biodiversity loss due to habitat loss and have to differentiate between logging types.

There are significant differences between the biodiversity effects of habitat loss and those of habitat degradation, and we want to emphasise that our MS examines the latter rather than the former. Habitat loss will always result in far more extreme, and more consistently negative, impacts on biodiversity than habitat degradation, which has been shown in previous meta-analyses (Gibson et al. 2011). This does not mean our results conflict with earlier work. Rather, it highlights the novelty of our work, in that we go beyond such black-and-white descriptions of land use change to treat that change as a continuous gradient.

Salvage logging is qualitatively different from logging for palm oil plantations. I missed more detailed information about the fate of the logged areas. The methods section only tells about a mix of fates, partly ending into palm oil plantations. The latter are known to retain a considerable biodiversity of birds if undisturbed forest is near. Salvage logging areas should also reduce biodiversity effects. In turn, the transformation of primary forest into open landscapes should have the most severe impact on diversity. Recalculating these different practices into a single metric, might miss this differences. From the methods section it did not became clear whether a differentiation into logging practices would be possible. However, for a paper that intends to provide guidelines I expected to see such reference to logging practice. In this respect a direct multitaxon comparison of primary forest and palm oil plantation might have been more informative.

Our goal in this MS was not to highlight the impacts of converting a forest to an oil palm plantation, which others have done (e.g. Savilaakso et al. 2014). Rather, we deliberately set out to understand how much biomass can be removed from a forest before it starts impacting forest biodiversity. To achieve our stated goal, we need to treat habitat degradation as a continuous gradient; if we were to categorise habitat changes in the way you suggest then we lose that ability. Our approach, then, is one in which we have attempted to harmonise into a single metric the various qualitative differences among logging practices that you rightly describe.

We also want to highlight that the qualitative categories of logging intensity that you describe do not align well with quantitative damage to ecosystems, for two reasons. First, many sites have been through multiple rounds of logging, with each round conducted at a different intensity and level of biomass extraction, and with a period of forest recovery in between. Apart from calling these areas “multiply logged”, it is difficult to categorise them clearly. Second, even within qualitative categories of logging intensity, there is a wide range of variation in the amount of biomass removed. We have previously quantified this at SAFE, and shown large overlaps in the standing biomass of forests that have been categorised as lightly logged, twice logged and salvage logged (see figure below, taken from Pfeifer et al. (2016)).

I also missed the temporal perspective. Biodiversity effects after habitat destruction often come with a time lag. Therefore, single annual surveys might not catch the silent biodiversity loss in time.

This is a good point and we agree with you. Time lags likely exist, but the complex history of our landscape makes this an intractable problem for analysis. Ideally, we could input time since disturbance as a predictor variable, but most of our study sites were logged multiple times over the past five to seven decades (Struebig et al. 2013, Riutta et al. 2018), so it is not clear how to quantify recovery time. The metric of habitat degradation that we use – biomass removed – provides a cumulative estimate of total degradation over the full history of land use in our study area. Your point about “silent biodiversity loss” also echoes that of Reviewer 2 who raised the issue of shifting baselines. In response to your and their comments, we have added a new section to the Methods describing the potential impact of time lags on our conclusions and highlighting the key assumptions our analysis makes in this regard (L709):

“Long-term shifts in the composition of forest communities might mean the biodiversity patterns we associate with primary forest in our data are themselves depauperate relative to historical patterns (Stouffer et al. 2021). Similarly, the complex logging history of our study site with repeated, but unequally distributed, rounds of logging means many sites have been through multiple stages of degradation separated by partial recovery (Struebig et al. 2013, Riutta et al. 2018). Our data are not sufficient to quantify historical patterns of occupancy nor the impact of time lags on trajectories of occupancy, so we are unable to directly test for these effects. Nonetheless, long term declines and local extinction of megafauna like the

Sumatran rhino *Dicerorhinus sumatrensis harrissoni* (Kretzschmar et al. 2016) make it likely that a shifting baseline is a valid concern at our study site. However, we have no way of knowing whether the rates of biodiversity change from the processes that might generate baseline shifts will be the same or different in primary and logged forest. Consequently, we can only emphasise that our analyses are based on a space-for-time substitution, which makes the implicit assumption that the effects of habitat degradation we quantify are additional to, and do not interact with, any other processes contributing to long-term biodiversity change.”

Much work on biodiversity loss after habitat reduction/transformation has focused on the species area relationship (SAR). The present degradation metric is related to the area variable of the SAR (biomass loss instead of area loss, both should be positively correlated). Given a power function SAR with slope of 0.5 (commonly found in small scale surveys) would retain 84% of species at 30% habitat loss. A slope of 0.3 predicts 90% species remaining. At 70% habitat loss and a slope of 0.5 only 55% of species would remain. That’s comparable to the present result. I think the relationship to the SAR approach needs to be discussed, maybe with pros and cons of both methods. How strong does biomass correlate with area?

This is an interesting observation. However, we are not focussed on species richness in this MS: none of our analyses use species richness as a response variable, which is the response variable in a SAR. We also note that this connection would require us to make a claim that habitat degradation at a point location can be linearly converted into a metric of habitat loss across an area. We believe making this connection would be speculative, and our data do not allow us to directly test this assumption. So while your calculations are intriguing, we believe such comparisons fall outside the scope of this MS.

Ewers et al. warn that single year snapshots might give highly inaccurate biodiversity patterns. This is surely true. The present paper tries to tackle this problem by the sheer mass of data and the assumption of statistical averaging. Nevertheless a quarter of data still relies on such single snapshots. All surveys were done in a single year. If these data were autocorrelated, for instance due to similar habitat and weather conditions during the respective study year, the results still would be biased. I think the authors need to clarify this point and present more information on when the surveys were done, whether multiple taxa were observed during such single surveys, and how large were the degrees of collinearity and of spatial autocorrelation.

Thank you – this is an important point for us to be clear about. Reviewer 2 also raised similar points and has asked for similar clarification. The responses we provided in our responses to them above simultaneously address your point here. In short, we have removed our claims that our data have overcome the challenge of context-dependence, and we have provided additional information about the distribution of repeat surveys, including the new Fig. S2.

For more than half of the taxa we analyse, data come from multiple years and are combined into a single model (either a GLMM or GLM). For those taxa, there is no single year that can be associated with the change or maximum rate points that we extract from those models, so we are unable to directly test for temporal autocorrelation in our results. However, in a separate MS examining the repeatability of results from these same data (Ewers et al. 2024), we have shown there is no signal of temporal autocorrelation: “There was no effect of the number of years between surveys on the probability of two surveys giving [the same] results (Fig S3A; binomial GLM: $\chi^2_{(1)} = 1.05$, $p = 0.30$).” We have made this secondary MS available to yourself and other readers on BioRxiv (Ewers et al. 2024), and added a new paragraph to the Methods of this present MS to describe this analysis (L700):

“Environmental conditions might influence the outcome of ecological studies (Powers and Hampton 2019). If the surveys we analyse here are unequally distributed through time, and taxon responses to habitat degradation are time-dependent, then temporal autocorrelation might influence our conclusions. In a separate analysis of the same data used in this study, we have quantified this effect and demonstrated it is not a concern (Ewers et al. 2024). We examined whether taxon-specific occurrence patterns across the habitat degradation gradient varied among surveys and years. We found that while occurrence patterns do vary among surveys, there was no consistent signal of survey year on those patterns. Specifically, the number of years between two surveys had no significant impact on the probability of two surveys reporting statistically indistinguishable response patterns.”

The results we present are derived from a single change point and single maximum rate point per taxon, and those points are not associated with a spatial location. Testing for spatial autocorrelation is therefore not relevant.

At the end of page 6 it is written “Of these, 1,681 taxa and all 126 functional groups were able to be modelled individually (≥ 5 occurrences).” I’m not sure what 5 occurrences means. Does this refer to 5 different sampling sites across the logging gradient? If yes how did you assess the sensitivity to logging

for such small sample sizes? In other words, would a different abundance cut-off give different results?

Apologies for the confusion. We have re-phrased this sentence to clarify our meaning (L233): “Of these, 1,681 taxa and all 126 functional groups were observed ≥ 5 times and were able to be modelled individually.”

In a separate MS based on the same dataset (Ewers et al. 2024), we have presented a sensitivity analysis demonstrating that a sample size of five is an appropriate cut-off. We provided this MS to the reviewers as a supporting document with our original submission, and have since uploaded it to BioRxiv. We have also added a specific statement to this effect in the Methods: “Sensitivity analyses on these same data have demonstrated that a cut-off of 5 occurrences is appropriate to generate consistently reliable results (Ewers et al. 2024)”

Beginning of line 7. Why is the area ‘experimental’? Are there artificially altered forest parts for scientific studies?

That is correct: there is an experimental design that has determined the spatial pattern of the most recent rounds of logging, and which is described in the cited reference (Ewers et al. 2024).

The data represent a wide number of taxa and functional types. However, they are still heavily biased with respect to biodiversity. Insects, spiders, nematodes, or molluscs are highly underrepresented, while the focus is on vertebrates and part of plants. This bias should have been discussed.

This is a good point, which we have tried to address in two ways. First, we present clear information about the number of taxa that were modelled and that provides quantitative insight into the magnitude of this potential taxonomic bias (L236): “The taxa were widely distributed across the tree of life (Fig. S1) and encompassed representatives from 86 taxonomic orders and 679 genera, including 590 plants (understorey and canopy, including grasses, herbs and woody trees), 88 mammals (volant and non-volant), 161 birds, 9 reptiles, 42 amphibians, 26 fish, and 635 invertebrates (including 263 beetles, 199 lepidopterans, 130 ants and 33 spiders) (Fig. S1).”

Second, the potential bias that you highlight would be an issue if the different taxa had notably different response patterns, but we have conducted several new analyses to

demonstrate that this is not the case. We used beta regression to examine differences in change points and maximum rate points with respect to taxon type, and compared both models to null models using log-likelihood tests. The results in both cases were non-significant (change points: $\chi^2_{(9)} = 2.79$, $P = 0.97$; maximum rate points: $\chi^2_{(-9)} = 9.78$, $P = 0.37$). We have added a new section to the Methods to present these tests and our conclusion that our main conclusions are not notably impacted by taxonomic bias (L691):

“While the taxa we examined were diverse and are widely distributed across the tree of life (Fig. S1), they are not evenly distributed across the tree of life. If the different taxa exhibit consistent variation in the pattern of their responses, this taxonomic bias might impact our overall conclusions. To test for this, we modelled both maximum rate points and change points as a function of taxonomic group, and used log-likelihood ratio tests to compare both models against a null model. There was no significant effect in either case (change points: $\chi^2_{(-9)} = 2.79$, $P = 0.97$; maximum rate points: $\chi^2_{(-9)} = 9.78$, $P = 0.37$), indicating taxonomic bias in our dataset is unlikely to influence the interpretation of our results.”

The caption of Fig. 3B is insufficient. The symbols are labelled as probability and severity, while the caption names susceptibility and sensitivity. Anyway the symbols needs clarification as captions should be self-explaining.

Apologies for this confusion, which Reviewer 1 also mentioned. We have updated both the symbology in Fig. 3 to be more easily interpretable, and the terminology in this caption to be consistent (L381).

References

- Brooks, T. M., R. A. Mittermeier, G. A. B. Da Fonseca, J. Gerlach, M. Hoffman, J. F. Lamoreux, C. G. Mittermeier, J. Pilgrim, and A. S. L. Rodrigues. 2006. Global biodiversity conservation priorities. *Science* **313**:58-61.
- Burivalova, Z., Ç. H. Şekerciöğlü, and L. P. Koh. 2014. Thresholds of logging intensity to maintain tropical forest biodiversity. *Current Biology* **24**:1893-1898.
- Carreño-Rocabado, G., M. Peña-Claros, F. Bongers, A. Alarcón, J.-C. Licona, and L. Poorter. 2012. Effects of disturbance intensity on species and functional diversity in a tropical forest. *Journal of Ecology* **100**:1453-1463.
- Ewers, R. M., W. D. Pearse, C. D. L. Orme, P. Amarasekare, T. D. Lorm, N. Granville, R. Adzhar, D. C. Aldridge, M. Ancrenaz, G. Atton, H. Barclay, M. L. Barclay, H. Bernard, J. E. Bicknell, T. R. Bishop, J. Blackman, S. Both, M. W. Boyle, H. Brant, E. Brasington, D. Burslem, E. R. Bush, K. Calloway, C. Carbone, L. Cator, P. M. Chapman, V. Chey, A.

- Chung, E. L. Clare, J. Cusack, M. Dancak, Z. G. Davies, C. W. Davison, M. M. Dawood, N. J. Deere, K. M. Dickinson, R. K. Didham, T. F. Dobert, R. A. Dow, R. Drinkwater, D. P. Edwards, P. Eggleton, A. Faruk, T. M. Fayle, A. H. Fikri, R. J. Fletcher, H. Folkard-Tapp, W. A. Foster, A. Fraser, R. Gill, R. J. Gray, R. Gray, N. Gregory, J. Hardwick, M. F. Harianja, J. K. Haysom, D. R. Hemprich-Bennett, S. P. Heon, M. Hrones, E. W. Jebrail, N. Jones, P. Jotan, V. A. Kemp, L. Kinneen, R. Kitching, O. Konopik, B. Kueh, I. Lane-Shaw, O. T. Lewis, S. H. Luke, E. Mackintosh, C. S. MacLean, N. Majalap, Y. Malhi, S. Martin, M. Massam, R. Matula, S. Maunsell, A. R. McKinlay, S. Mitchell, K. E. Mullin, R. Nilus, C. D. Noble, J. M. Parrett, M. Pfeifer, A. Pianzin, L. Picinali, R. Pillay, F. Poznansky, A. Prairie, L. Qie, H. Rahman, T. Riutta, S. J. Rossiter, J. M. Rowcliffe, G. B. Roxby, D. I. Seaman, S. S. Sethi, A. Shabrani, A. Sharp, E. M. Slade, J. Sleutel, N. Stork, M. Struebig, M. Svatek, T. Swinfield, H. H. Tan, Y. A. Teh, J. Thorley, E. C. Turner, J. P. Twining, M. Vollans, O. Wearn, B. L. Webber, F. Wiederkehr, C. L. Wilkinson, J. Williamson, A. Wong, D. J. Yeo, N. Yoh, K. M. Yusah, G. Yvon-Durocher, N. Zulkifli, O. Z. Daniel, G. Reynolds, and C. C. Banks-Leite. 2024. Variable responses of individual species to tropical forest degradation. *bioRxiv:2024.2002.2009.576668*.
- Gibson, L., T. M. Lee, L. P. Koh, B. W. Brook, T. A. Gardner, J. Barlow, C. A. Peres, C. J. A. Bradshaw, W. F. Laurance, T. E. Lovejoy, and N. S. Sodhi. 2011. Primary forests are irreplaceable for sustaining tropical biodiversity. *Nature* **478**:378-381.
- Kretzschmar, P., S. Kramer-Schadt, L. Ambu, J. Bender, T. Bohm, M. Ernsing, F. Göritz, R. Hermes, J. Payne, N. Schaffer, S. T. Thayaparan, Z. Z. Zainal, T. B. Hildebrandt, and H. Hofer. 2016. The catastrophic decline of the Sumatran rhino (*Dicerorhinus sumatrensis harrissoni*) in Sabah: Historic exploitation, reduced female reproductive performance and population viability. *Global Ecology and Conservation* **6**:257-275.
- Malhi, Y., T. Riutta, O. R. Wearn, N. J. Deere, S. L. Mitchell, H. Bernard, N. Majalap, R. Nilus, Z. G. Davies, R. M. Ewers, and M. J. Struebig. 2022. Logged tropical forests have amplified and diverse ecosystem energetics. *Nature* **612**:707-713.
- Orme, C. D. L., S. Mayor, L. dos Anjos, P. F. Develey, J. H. Hatfield, J. C. Morante-Filho, J. M. Tylianakis, A. Uezu, and C. Banks-Leite. 2019. Distance to range edge determines sensitivity to deforestation. *Nature Ecology & Evolution* **3**:886-891.
- Pfeifer, M., L. Kor, R. Nilus, E. Turner, J. Cusack, I. Lysenko, M. Khoo, V. K. Chey, A. C. Chung, and R. M. Ewers. 2016. Mapping the structure of Borneo's tropical forests across a degradation gradient. *Remote Sensing of Environment* **176**:84-97.
- Pfeifer, M., V. Lefebvre, E. Turner, J. Cusack, M. S. Khoo, V. K. Chey, M. Peni, and R. M. Ewers. 2015. Deadwood biomass: an underestimated carbon stock in degraded tropical forests? *Environmental Research Letters* **10**:044019.
- Pinard, M. A., and F. E. Putz. 1996. Retaining forest biomass by reducing logging damage. *Biotropica* **28**:278-295.
- Powers, S. M., and S. E. Hampton. 2019. Open science, reproducibility, and transparency in ecology. *Ecological Applications* **29**:e01822.
- R Development Core Team. 2021. R: A language and environment for statistical computing. R Foundation for Statistical Computing, Vienna.
- Reynolds, G., J. Payne, W. Sinun, G. Mosigil, and R. P. D. Walsh. 2011. Changes in forest land use and management in Sabah, Malaysian Borneo, 1990-2010, with a focus on the Danum Valley region. *Philosophical Transactions of the Royal Society B-Biological Sciences* **366**:3168-3176.
- Riutta, T., Y. Malhi, L. K. Kho, T. R. Marthews, W. Huaraca Huasco, M. Khoo, S. Tan, E. Turner, G. Reynolds, S. Both, D. F. R. P. Burslem, Y. A. Teh, C. S. Vairappan, N. Majalap, and R.

- M. Ewers. 2018. Logging disturbance shifts net primary productivity and its allocation in Bornean tropical forests. *Global Change Biology* **24**:2913-2928.
- Rosoman, G., S. S. Sheun, C. Opal, P. Anderson, and R. Trapshah. 2017. The HCS Approach Toolkit Version 2.0. HCS Approach Steering Group, Singapore.
- Savilaakso, S., C. Garcia, J. Garcia-Ulloa, J. Ghazoul, M. Groom, M. R. Guariguata, Y. Laumonier, R. Nasi, G. Petrokofsky, J. Snaddon, and M. Zrust. 2014. Systematic review of effects on biodiversity from oil palm production. *Environmental Evidence* **3**:4.
- Schultz, B., D. Brockington, E. A. Coleman, I. Djenontin, H. W. Fischer, F. Fleischman, P. Kashwan, K. Marquardt, M. Pfeifer, R. Pritchard, and V. Ramprasad. 2022. Recognizing the equity implications of restoration priority maps. *Environmental Research Letters* **17**:114019.
- Stouffer, P. C., V. Jirinec, C. L. Rutt, R. O. Bierregaard Jr, A. Hernández-Palma, E. I. Johnson, S. R. Midway, L. L. Powell, J. D. Wolfe, and T. E. Lovejoy. 2021. Long-term change in the avifauna of undisturbed Amazonian rainforest: ground-foraging birds disappear and the baseline shifts. *Ecology Letters* **24**:186-195.
- Struebig, M., A. Turner, E. A. Giles, F. Lasmana, S. Tollington, H. Bernard, and D. Bell. 2013. Quantifying the biodiversity value of repeatedly logged rainforests: gradient and comparative approaches from Borneo. *Advances in Ecological Research* **48**:183-224.
- Williams, J. J., and T. Newbold. 2021. Vertebrate responses to human land use are influenced by their proximity to climatic tolerance limits. *Diversity and Distributions*.

Reviewer Reports on the First Revision:

Referees' comments:

Referee #1 (Remarks to the Author):

Thank you to the authors for taking the time to respond to all my concerns and questions, and for amending the text and figures where necessary. I have no further comment to make, but would like to congratulate the authors again on this impressive work.

Referee #2 (Remarks to the Author):

This is the second time I am looking at this manuscript -- I reviewed the original submission and now have read through the revised version as well as the responses to reviewer comments. This study examines logging thresholds for a wide variety of taxonomic groups from Sabah, Malaysia.

The authors have made a great effort to address the the editor and reviewers' concerns. I am satisfied with the revised manuscript and congratulate the authors on a fine piece of work. The data are impressive, the work has great significance for conservation policy, the analytical approach and conclusions are robust.

I do not have any further comments on this manuscript.

Referee #3 (Remarks to the Author):

In their resubmission Ewers et al. provide detailed responses to all of the issues raised by the referees. Particularly, they show that collinearity did not influence the results. Referee 1 pointed to recovery times. The authors discuss this issue in the methods section. Mentioning this critical point in the main text, too, would improve the paper. Differential outcomes due to recovery times cannot be avoided and I think the use of averaged values is appropriate in the present case.

In general, the manuscript has improved by new text answering the comments and questions of the referees. However, I must say that in the light of the answers (also to the comments of the other referees) I got some more doubts about the generality of the results. Given the economic and conservation implication this might be a critical point. The new version now mentions these issues. However, it should be made very clear that the present results refer to a case study where local conditions might influence the outcome (tipping points might be very variable depending on local conditions). Further, the paper refers to a specific type of degradation of primary forest. Diversity effects due to altered tree composition in secondary or managed forests (many African forests) are not dealt with. Long-term effects are also not considered

The results rely on the specific way of calculating the two points of change. The function of the occurrence model must be differentiable at least three times. Therefore, data were squeezed into the specific binomial model with the basic assumption that occurrence follows a type III functional response (similar to a tipping point model). Other possible response functions are discarded even if

they would fit better. Additionally, AIC was used to choose the appropriate spatial scale. Scale is important here but the model should focus on precision (fit) not on information content. I wonder whether the results remain stable after altering scale assignment.

The paper studies habitat degradation and in the answer to my question author admit that habitat loss might have a far more negative impact on biodiversity. Here, I wish to take a different perspective. Tropical forests harbour many specialist species associated to single microhabitats or specific feeding plants. Salvage logging might degrade the forest, but for many of such species this would be equivalent to habitat destruction (loss). I admit that it will be nearly impossible to assess this effect. Nevertheless it might influence interpretation.

I still think that the present approach is analogue to the species – area relationship. Biomass removal (or degradation) can be seen as a reduction of available space to live and this space is (often) allometrically related to richness (here quantified as occurrence probability). Allometric functions do not have these nice inflection points but might otherwise be well suited to model species occurrences and richness. Did you compare Type I and II occurrence responses with the type III response used here?

##Additional comment##

I could not check one additional concern but in my feeling the two points of change identified here seem to be not independent. I guess that the position of the change point constraints the position of the maximum rate point.

Author Rebuttals to First Revision: Referee #3 (Remarks to the Author):

In their resubmission Ewers et al. provide detailed responses to all of the issues raised by the referees. Particularly, they show that collinearity did not influence the results. Referee 1 pointed to recovery times. The authors discuss this issue in the methods section. Mentioning this critical point in the main text, too, would improve the paper. Differential outcomes due to recovery times cannot be avoided and I think the use of averaged values is appropriate in the present case.

This is a good point. We have now moved some text from the Methods to the main MS to make it clear that our metric of forest degradation is a balance between the processes of logging and forest recovery time (Line 242):

“Along this gradient, the percentage of biomass removed varied from zero through to 99 %, which we use as a generalised metric of forest degradation. This metric implicitly combines the initial removal of woody biomass through logging and land clearance with the gradual recovery of biomass that may have occurred since the last disturbance event(s), meaning our metric of forest degradation reflects the present-day balance between these two opposing forces.”

In general, the manuscript has improved by new text answering the comments and questions of the referees.

Thank you. We're pleased the revisions have made a positive difference.

However, I must say that in the light of the answers (also to the comments of the other referees) I got some more doubts about the generality of the results. Given the economic and conservation implication this might be a critical point. The new version now mentions these issues. However, it should be made very clear that the present results refer to a case study where local conditions might influence the outcome (tipping points might be very variable depending on local conditions).

As you point out, we already raise these issues in the main text. We specifically highlight the fact that our study is conducted at a single site, and also cite literature that might help predict exactly *how* tipping points might vary among locations. We've now slightly expanded that section (L435):

“Our data were collected from a single site, however, and taxon responses to habitat degradation can vary across geographical gradients (Orme et al. 2019, Williams and Newbold 2021) meaning the exact location of taxon-specific thresholds might similarly vary, so more studies of a similar nature will be required to strengthen confidence in the generality of our conclusions.”

Further, the paper refers to a specific type of degradation of primary forest. Diversity effects due to altered tree composition in secondary or managed forests (many African forests) are not dealt with.

The composition of tree communities varies along logging gradients in SE Asia – including the one we examine (Both et al. 2019) – just as it does in African forests. This is an important point to note, so we have now added text to the main MS to highlight the range of environmental conditions that change along the logging gradient, of which tree composition is just one example (L247):

“From previous work, we have shown that forest degradation causes changes to local environmental conditions, including the microclimate (Hardwick et al. 2015) and the functional composition of the tree community (Both et al. 2019).”

Long-term effects are also not considered

You are right to raise this concern, as did your co-reviewers in the previous round. In response we provided a paragraph that specifically addresses the issues of how our results might be impacted by long-term time lags (L747):

*“Long-term shifts in the composition of forest communities might mean the biodiversity patterns we associate with primary forest in our data are themselves depauperate relative to historical patterns (Stouffer et al. 2021). Similarly, the complex logging history of our study site with repeated, but unequally distributed, rounds of logging means many sites have been through multiple stages of degradation separated by partial recovery (Struebig et al. 2013, Riutta et al. 2018). Our data are not sufficient to quantify historical patterns of occupancy nor the impact of time lags on trajectories of occupancy, so we are unable to directly test for these effects. Nonetheless, long term declines and local extinction of megafauna like the Sumatran rhino *Dicerorhinus sumatrensis harrissoni* (Kretzschmar et*

al. 2016) make it likely that a shifting baseline is a valid concern at our study site. However, we have no way of knowing whether the rates of biodiversity change from the processes that might generate baseline shifts will be the same or different in primary and logged forest. Consequently, we can only emphasise that our analyses are based on a space-for-time substitution, which makes the implicit assumption that the effects of habitat degradation we quantify are additional to, and do not interact with, any other processes contributing to long-term biodiversity change.”

The results rely on the specific way of calculating the two points of change. The function of the occurrence model must be differentiable at least three times.

We only need to differentiate the occurrence model two times – not three. The differentiated functions are demonstrated in Fig. S3.

Therefore, data were squeezed into the specific binomial model with the basic assumption that occurrence follows a type III functional response (similar to a tipping point model). Other possible response functions are discarded even if they would fit better.

The data we analyse are presence-absence data, and binomial models are the appropriate statistical modelling framework for analysing data of this nature. Other models and response functions, which you suggest below, have inappropriate error structures for handling binomial data.

Additionally, AIC was used to choose the appropriate spatial scale. Scale is important here but the model should focus on precision (fit) not on information content. I wonder whether the results remain stable after altering scale assignment.

We have used an accepted, standard statistical framework to explain the patterns in our data. At no point do we use our analysis to try and predict occurrence probabilities for taxa at unobserved sites. Had we attempted to do this, then we agree that a focus on precision would be more appropriate. However, because our focus is entirely on describing patterns, an analysis focus on the information content of fitted statistical models is the most appropriate framework to use.

Our approach to identifying the taxon-specific scale for analysis mirrors approaches widely used in the landscape ecology literature for many years (e.g. Steffan-Dewenter et

al. 2002, Umetsu et al. 2008, Banks-Leite et al. 2011), and as such we believe it is a robust one.

The paper studies habitat degradation and in the answer to my question author admit that habitat loss might have a far more negative impact on biodiversity. Here, I wish to take a different perspective. Tropical forests harbour many specialist species associated to single microhabitats or specific feeding plants. Salvage logging might degrade the forest, but for many of such species this would be equivalent to habitat destruction (loss). I admit that it will be nearly impossible to assess this effect. Nevertheless it might influence interpretation.

We disagree with the definition of the term “habitat” that you are using here. In our MS, we used this term in relation to the forest as a whole, which is consistent with common use of the term in the conservation and ecological literature that examines the impacts of habitat loss on biodiversity (e.g. Pimm and Askins 1995, Wearn et al. 2012, Betts et al. 2017). These high-profile papers, and many more, routinely use ‘forest’ and ‘habitat’ interchangeably. By contrast, your comment implements a taxon-specific definition in which “habitat” is the combination of environmental conditions and feeding opportunities exploited by a given taxon. We interpret this as the “niche” of a species rather than their “habitat,” as your description is consistent with Hutchinson’s (1957) classic work on this topic.

We certainly agree that habitat degradation – using our definition of the term – will result in the loss of specific niches for some species. This is a key mechanism by which habitat degradation exerts impacts on biodiversity. To emphasise this, we have added text to the MS to highlight how forest degradation drives changes to the environmental conditions of the habitat (L247):

“From previous work, we have shown that forest degradation causes changes to local environmental conditions, including the microclimate (Hardwick et al. 2015) and the functional composition of the tree community (Both et al. 2019).”

I still think that the present approach is analogue to the species – area relationship. Biomass removal (or degradation) can be seen as a reduction of available space to live and this space is (often) allometrically related to richness (here quantified as occurrence probability).

We do not analyse species richness in this MS, and as such any analogue with the species-area relationship is taking our results out of context. You are also claiming that our metric of occurrence probability is somehow analogous to species richness, but this is not the case. Our metric is calculated individually for each of the taxa we analyse, whereas richness is an aggregated metric across multiple taxa. There is a considerable difference in statistical treatment of these two data types: occurrence probabilities are based on binary data and have a binomial error structure, whereas species richness is count data with a poisson error structure.

Allometric functions do not have these nice inflection points but might otherwise be well suited to model species occurrences and richness. Did you compare Type I and II occurrence responses with the type III response used here?

Respectfully, we think you are confusing the sigmoid shape of a binomial GLM with Holling's (1963) classic Type I, II and III functional response curves. We are modelling the occurrence patterns of taxa, whereas functional response curves model the consumption rates of taxa, and these are not interchangeable concepts.

Our data represent presence or absence of a given taxon in a survey, and hence are a binary response variable. Binomial GLMs are appropriate for modelling these data, as predictions of the model are also correctly bounded in (0,1): the probability of observing a species cannot be lower than zero or higher than one. By contrast, Holling's Type II and III models have a lower bound at zero, but their asymptote is at the maximum consumption rate which may be greater than one; and the predictions for Type I responses are completely unbounded. Type I, II and III functional response curves are fitted using linear or non-linear regression with Gaussian errors, but these methods do not correctly account for the error distribution of binomial data.

##Additional comment##

I could not check one additional concern but in my feeling the two points of change identified here seem to be not independent. I guess that the position of the change point constraints the position of the maximum rate point.

This is correct: the maximum rate point is constrained to occur at a higher level of biomass loss than the change point. This is an ecologically appropriate constraint, as any given taxon must begin responding to biomass loss (change point) *before* the rate of that response can accelerate to its maximum (maximum rate point).

References

- Banks-Leite, C., R. M. Ewers, V. Kapos, A. C. Martensen, and J. P. Metzger. 2011. Comparing species with measures of landscape structure as indicators of ecological integrity. *Journal of Applied Ecology* **48**:706-714.
- Betts, M. G., C. Wolf, W. J. Ripple, B. Phalan, K. A. Millers, A. Duarte, S. H. M. Butchart, and T. Levi. 2017. Global forest loss disproportionately erodes biodiversity in intact landscapes. *Nature* **547**:441-444.
- Both, S., T. Riutta, C. E. T. Paine, D. M. O. Elias, R. S. Cruz, A. Jain, D. Johnson, U. H. Kritzler, M. Kuntz, N. Majalap-Lee, N. Mielke, M. X. Montoya Pillco, N. J. Ostle, Y. Arn Teh, Y. Malhi, and D. F. R. P. Burslem. 2019. Logging and soil nutrients independently explain plant trait expression in tropical forests. *New Phytologist* **221**:1853-1865.
- Hardwick, S. R., R. Toumi, M. Pfeifer, E. C. Turner, R. Nilus, and R. M. Ewers. 2015. The relationship between leaf area index and microclimate in tropical forest and oil palm plantation: forest disturbance drives changes in microclimate. *Agricultural and Forest Meteorology* **201**:187-195.
- Holling, C. S. 1963. The functional response of predators to prey density and its role in mimicry and population regulation. *Memoirs of the Entomological Society of Canada* **97**:5-60.
- Hutchinson, G. 1957. Concluding remarks. *Cold Springs Harbour Symposia on Quantitative Biology* **22**:415-427.
- Kretzschmar, P., S. Kramer-Schadt, L. Ambu, J. Bender, T. Bohm, M. Ernsing, F. Göritz, R. Hermes, J. Payne, N. Schaffer, S. T. Thayaparan, Z. Z. Zainal, T. B. Hildebrandt, and H. Hofer. 2016. The catastrophic decline of the Sumatran rhino (*Dicerorhinus sumatrensis harrissoni*) in Sabah: Historic exploitation, reduced female reproductive performance and population viability. *Global Ecology and Conservation* **6**:257-275.
- Orme, C. D. L., S. Mayor, L. dos Anjos, P. F. Develey, J. H. Hatfield, J. C. Morante-Filho, J. M. Tylianakis, A. Uezu, and C. Banks-Leite. 2019. Distance to range edge determines sensitivity to deforestation. *Nature Ecology & Evolution* **3**:886-891.
- Pimm, S. L., and R. A. Askins. 1995. Forest losses predict bird extinctions in eastern North America. *Proceedings of the National Academy of Sciences* **92**:9343-9347.
- Riutta, T., Y. Malhi, L. K. Kho, T. R. Marthews, W. Huaraca Huasco, M. Khoo, S. Tan, E. Turner, G. Reynolds, S. Both, D. F. R. P. Burslem, Y. A. Teh, C. S. Vairappan, N. Majalap, and R. M. Ewers. 2018. Logging disturbance shifts net primary productivity and its allocation in Bornean tropical forests. *Global Change Biology* **24**:2913-2928.
- Steffan-Dewenter, I., U. Münzenburg, C. Bürger, C. Thies, and T. Tschardtke. 2002. Scale-dependent effects of landscape context on three pollinator guilds. *Ecology* **83**:1421-1432.
- Stouffer, P. C., V. Jirinec, C. L. Rutt, R. O. Bierregaard Jr, A. Hernández-Palma, E. I. Johnson, S. R. Midway, L. L. Powell, J. D. Wolfe, and T. E. Lovejoy. 2021. Long-term change in the

- avifauna of undisturbed Amazonian rainforest: ground-foraging birds disappear and the baseline shifts. *Ecology Letters* **24**:186-195.
- Struebig, M., A. Turner, E. A. Giles, F. Lasmana, S. Tollington, H. Bernard, and D. Bell. 2013. Quantifying the biodiversity value of repeatedly logged rainforests: gradient and comparative approaches from Borneo. *Advances in Ecological Research* **48**:183-224.
- Umetsu, F., J. P. Metzger, and R. Pardini. 2008. Importance of estimating matrix quality for modeling species distribution in complex tropical landscapes: a test with Atlantic forest small mammals. *Ecography* **31**:359-370.
- Wearn, O. R., D. C. Reuman, and R. M. Ewers. 2012. Extinction debt and windows of conservation opportunity in the Brazilian Amazon. *Science* **337**:228-232.
- Williams, J. J., and T. Newbold. 2021. Vertebrate responses to human land use are influenced by their proximity to climatic tolerance limits. *Diversity and Distributions*.

Reviewer Reports on the Second Revision:

Referees' comments:

Referee #3 (Remarks to the Author):

This revised version clarified most of my comments and questions.

The main text now contains comments and references explaining to which degree the paper can be seen as a case study and whether forest specificity might influence the results. The question of non-independence of the change and maximum rate point might have been explained a bit more detailed but this is a minor point. I'm sure the paper will raise controversy.